# Age-associated B cells predict impaired humoral immunity after COVID-19 vaccination in patients receiving immune checkpoint blockade

Age-associated B cells (ABC) accumulate with age and in individuals with different immunological disorders, including cancer patients treated with immune checkpoint blockade and those with inborn errors of immunity. Here, we investigate whether ABCs from different conditions are similar and how they impact the longitudinal level of the COVID-19 vaccine response. Single-cell RNA sequencing indicates that ABCs with distinct aetiologies have common transcriptional profiles and can be categorised according to their expression of immune genes, such as the autoimmune regulator (*AIRE*). Furthermore, higher baseline ABC frequency correlates with decreased levels of antigen-specific memory B cells and reduced neutralising capacity against SARS-CoV-2. ABCs express high levels of the inhibitory FcγRIIB receptor and are distinctive in their ability to bind immune complexes, which could contribute to diminish vaccine responses either directly, or indirectly via enhanced clearance of immune complexed-antigen. Expansion of ABCs may, therefore, serve as a biomarker identifying individuals at risk of suboptimal responses to vaccination.

Despite the success of mRNA-lipid nanoparticle (mRNA-LNP) COVID-19 vaccines in reducing the risk of symptomatic infection, hospitalisation, and death[1,2], vaccinated patients with cancer remain at increased risk of severe outcomes following SARS-CoV-2 infection[3]. Immune checkpoint blockade (ICB) is a cancer therapy that observational studies[4–6] suggest could improve the efficacy of vaccines. By targeting PD-1 and CTLA-4 checkpoints, ICB non-specifically promotes T cell responses, including those involved in anti-cancer and anti-viral immunity. Furthermore, preclinical evidence indicates immune checkpoint blockade can, via T and B cell interactions, enhance antibody responses[7,8]. The neutralising antibodies produced by the humoral response are a vital component of vaccine protection as they inhibit viral replication in vitro and correlate with protection against infection in vivo[9–11]. However, complicating the positive potential of ICB for vaccine enhancement, ICB induces the expansion of a B cell subset termed in other contexts Age-Associated B cells (ABC)[12], which may have a confounding effect on humoral vaccine responses.

ABCs are a naturally occurring population of antigen-experienced B cells which expands continuously with age in healthy individuals but accumulates prematurely in people with certain immune dyscrasias, autoimmunity, and/or infectious diseases[13]. They have also been termed CD21[low] B cells, CD11c[+] B cells, CD11c[+]T-bet[+] B cells, double negative 2 (DN2) B cells or atypical memory B cells[13–20], and may additionally be generated by external challenges[21,22], including COVID-19 infection and vaccination[23]. Although ABCs are associated with disease in autoimmunity, their role in vaccine immunity is uncertain. In mice, ABCs are required for optimal antibody responses following influenza vaccination[24], possibly due to a greater ability than follicular B cells to present antigens to T cells[16,25]. In patients with cancer treated with ICB, the expansion of ABCs precedes the development of both

✉ e-mail: jcy28@cam.ac.uk; jedt2@cam.ac.uk

antibody-mediated and non-antibody-mediated autoimmunity[12]. Furthermore, we and others have shown that expansion of ABCs is associated with antibody deficiency in specific cohorts of patients with inborn errors of immunity (IEI)[26–30]. These include patients with *NFKB1* haploinsufficiency, in whom the genetic lesion leads to a combined B and T cell defect[29], and patients with *CTLA4* haploinsufficiency, in whom the genetic-lesion leads predominantly to a B cell-extrinsic functional T cell deficit[27,31]. These observations suggest two key questions: are the ABCs found in patients treated with ICB equivalent to those found in other settings? And do these cells, when expanded through distinct mechanisms, have a positive or negative effect on humoral vaccine responses?

The increase in patients with cancer eligible to receive ICB treatment[32] and the survival benefit conferred by ICB treatment means that optimally protecting patients with cancer who are receiving these therapies from infection will be an increasing priority.

Here, we investigate the characteristics and activity of ABCs in different patient groups and explore the interplay between ABCs and successful vaccination. We show that ABCs across distinct aetiologies exhibit common transcriptional profiles and that a high frequency of ABC before vaccination is associated with decreased levels of antigen-specific memory B cells, and diminished neutralising capacity against SARS-CoV-2. Additionally, we identify increased expression of the inhibitory receptor FcγRIIB on ABCs as a possible factor contributing to the reduced response to vaccination in individuals with elevated ABC levels.

## Results

### ABCs arising from distinct aetiologies have common transcriptional profiles

We first assessed the similarities between ABCs from patients with different causes for ABC expansion, using single cell RNA sequencing (scRNAseq). This included B cells from healthy controls (HC, $n = 8$), patients with cancer treated with ICB (ICB, $n = 8$) and patients with *NFKB1* or *CTLA4* haploinsufficiency (IEI, $n = 4$), together with a published set of single B cell transcriptomes from patients with systemic lupus erythematosus (SLE, $n = 3$)[33]. By focussing on patients with rare, monogenic defects leading to well-characterised IEI, we were able to ensure that contrasting B cell intrinsic (NFKB1) and extrinsic (CTLA4) aetiologies were included. Unsupervised clustering of the 52,402 B cells, displayed in a uniform manifold approximation and projection (UMAP) visualisation, showed six clusters (Fig. 1a). Using CITE-seq, we identified ABCs that had low CD21 and high CD11c surface protein expression, in keeping with the definitions used in some previous flow cytometry-based studies[16,34] (Supplementary Fig. 1a, b). Cells were dispersed across all clusters regardless of the patient group or sex from which they originated (Fig. 1b–d). ABC frequencies were higher in all patient groups compared to healthy controls (Fig. 1d).

### Distinct subsets of ABCs express genes associated with distinct immune functions

The clustering and marker gene expression suggested that the ABC population can be subdivided into "Classical ABCs", "Anergic ABCs" and "CD1c ABCs" (Fig. 1e). Classical ABC frequencies were higher in females and patients with SLE (Fig. 1b–d and Supplementary Fig. 1c–f). ABCs have been described as "anergic B cells"[35,36]. Interestingly, anergic ABCs (expressing canonical genes associated with anergy, such as *EGR1* and *NR4A1*) clustered separately from Classical ABCs (Fig. 1a, e and Supplementary Data File 1), indicating heterogeneity within ABCs for this phenotype. Indeed, many of the most differentially expressed genes in Classical ABCs were associated with MHC class II-restricted antigen presentation (Fig. 1f). Gene ontology enrichment analysis demonstrated selective upregulation of biological processes associated with professional antigen presentation, such as antigen uptake, processing, class II presentation, and co-stimulation (Fig. 1g and

Supplementary Table 1). This was specific for MHC class-II-restricted antigen presentation, as genes required for MHC class I-restricted antigen presentation were instead differentially expressed in CD1c ABCs, relative to Classical ABCs (Supplementary Fig. 2). This supports the functional separation of these two clusters, which were less distinct on UMAP visualisation (Fig. 1a). To further analyse these different clusters, we used single cell BCR sequencing data to calculate the vertex/node Gini index as a measure of clonal expansion[37] and the degree index from dandelion[38,39] as a measure of how cells are clonally related. The classical ABC cluster had the highest Gini index and second highest degree index, suggesting that more clones of classical ABCs are expanded and that these cells are clonally related (Supplementary Fig. 3).

### Upregulation of AIRE and its target genes is seen in classical ABCs

Consistent with our gene ontology analysis, we confirmed MHC-II antigen presentation related genes, such as HLA-proteins, *CD74* and *CD86*, were amongst the most differentially expressed genes of Classical ABCs. Additionally, other important genes involved in the processing of peptides, oligosaccharides and fatty acids such as *LGMN, IFI30, PSAP* and *ASAH1* were differentially expressed (Fig. 2a and Supplementary Data File 2). Intriguingly, this analysis also revealed the Autoimmune Regulator (*AIRE*) to be significantly upregulated in Classical ABCs from all participant cohorts (Fig. 2a, b, Supplementary Fig. 4). *AIRE*-expressing B lymphocytes were also in other clusters, albeit at a much lower frequency (Fig. 2c). This is consistent with the accumulation of *AIRE*-expressing B cells in the Classical ABC differentiation state during peripheral B cell development, rather than selective upregulation of *AIRE* after ABC differentiation. Expression of a single gene in 3.85% of cells within a subset will make a negligible contribution to the UMAP clustering, therefore enrichment of *AIRE*-expressing cells within the apex of the Classical ABC cluster (Supplementary Fig. 4a) further supports the biological association of *AIRE* expression and ABC differentiation. AIRE is a transcriptional regulator expressed in both the thymic epithelium and thymic B cells[40,41], which increases expression of tissue-restricted antigens (TRAs). TRA expression in the thymus is a central tolerance mechanism, whereby stimulation of developing T cells by antigens usually expressed in non-thymic tissues directs these cells away from fates with the potential for harmful self-antigen driven autoimmune responses. Previous reports have suggested that AIRE is not functional in the small proportion of peripheral lymphocytes in which it can be detected[42–44], although AIRE function has never been assessed in ABCs (a peripheral B cell subset specifically associated with autoimmune disease[16,45,46]). *AIRE*-expressing cells also upregulated a set of genes previously defined as AIRE targets in thymic B cells[41], which are predominantly expressed in other tissues such as brain (Fig. 2d, Supplementary Table 2 and Supplementary Fig. 5). In addition, Classical ABCs upregulated HLA-G (Supplementary Fig. 2), a non-classical class I HLA molecule transactivated by AIRE in thymic epithelial cells[47]. Taken together, these data provide functional evidence of *AIRE* expression in Classical ABCs. Further work will be necessary to evaluate if *AIRE* expression in ABCs leads to self-antigen expression and productive presentation which can initiate autoimmune responses.

### ABC frequency predicts neutralising antibody response to COVID-19 vaccination

To investigate the impact of ABCs on mRNA-LNP vaccination, we next analysed the immune response to the mRNA BNT162b2 COVID-19 vaccine in patients with variable expansion of ABCs, including healthy controls (HC, $n = 10$), patients with cancer treated with ICB (ICB, $n = 19$) and patients with *NFKB1* or *CTLA4* haploinsufficiency (IEI, $n = 9$). One donor with phenotypic similarities to patients with CTLA4 deficiency who remained genetically unclassified was also included (Fig. 3a and

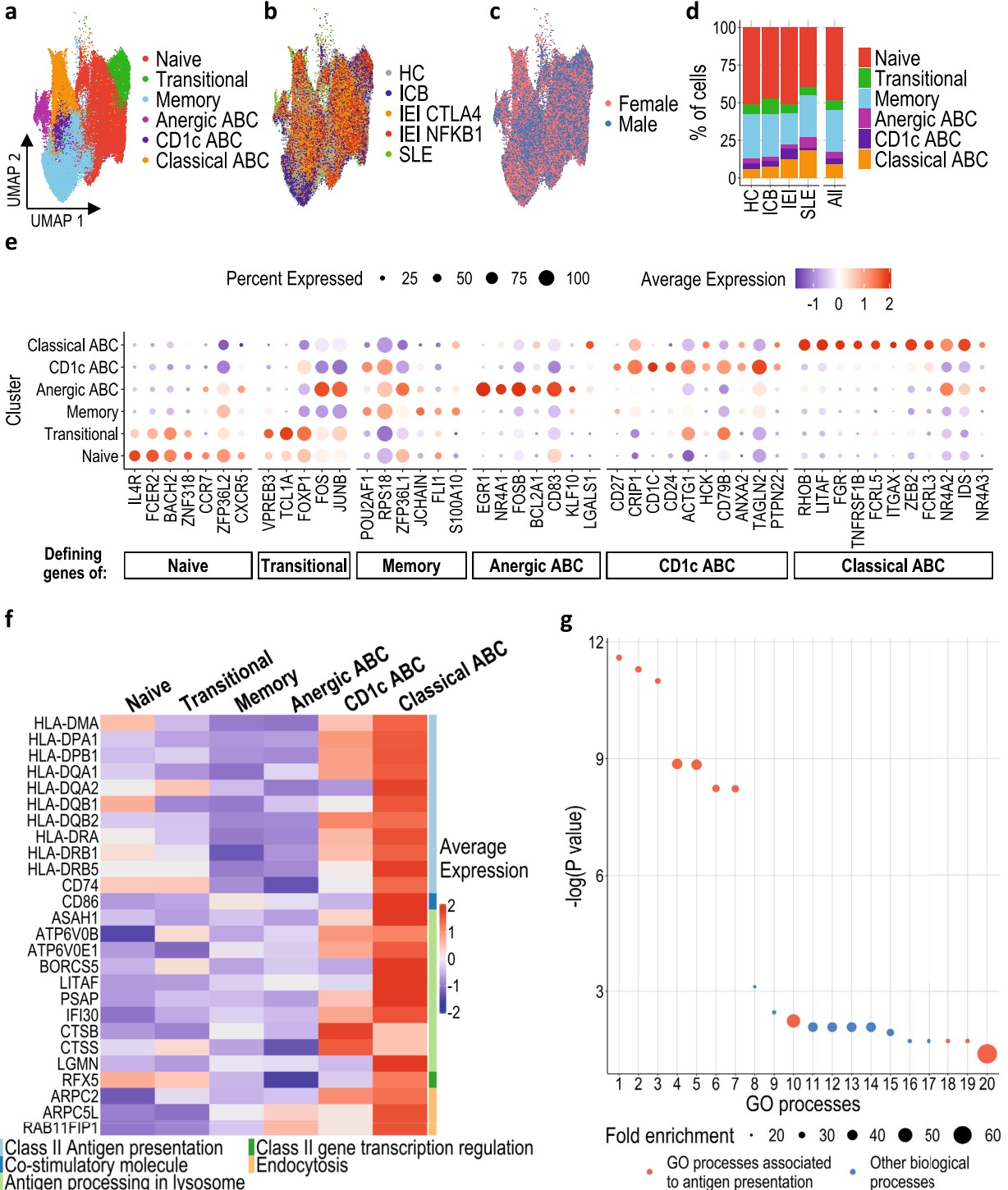

**Fig. 1 | Single cell transcriptional signature of ABCs.** Magnetically enriched B cells from PBMCs were analysed through droplet-based single cell RNA sequencing technology (n = 52402). UMAPs of cells from all individuals coloured by (**a**) annotated Louvain clusters, (**b**) health condition (HC healthy controls, ICB immune-checkpoint blockade treated cancer patients, IEI CTLA-4/NFKB1, inborn errors of immunity, CTLA-4 and NFKB1 mutants respectively, SLE systemic lupus erythematosus patients), and (**c**) gender. Dots represent a single cell. **d** Proportion of total cells from each health condition belonging to each cluster, clusters coloured as indicated. **e** Expression of 46 genes in each B cell cluster which define the different subpopulations. Dot size represents proportion of cells within a cluster expressing the indicated gene and colours represent the average expression level. **f** Heatmap representing scaled expression values of genes associated with antigen uptake, processing and class-II presentation in each cluster. **g** Gene ontology enrichment analysis of biological processes associated with upregulated genes in classical ABCs. GO terms with highest fold enrichments are ranked by -log (P value). Statistical testing via Fisher's Exact test with correction for false discovery rate. Dot size is proportional to the fold enrichment.

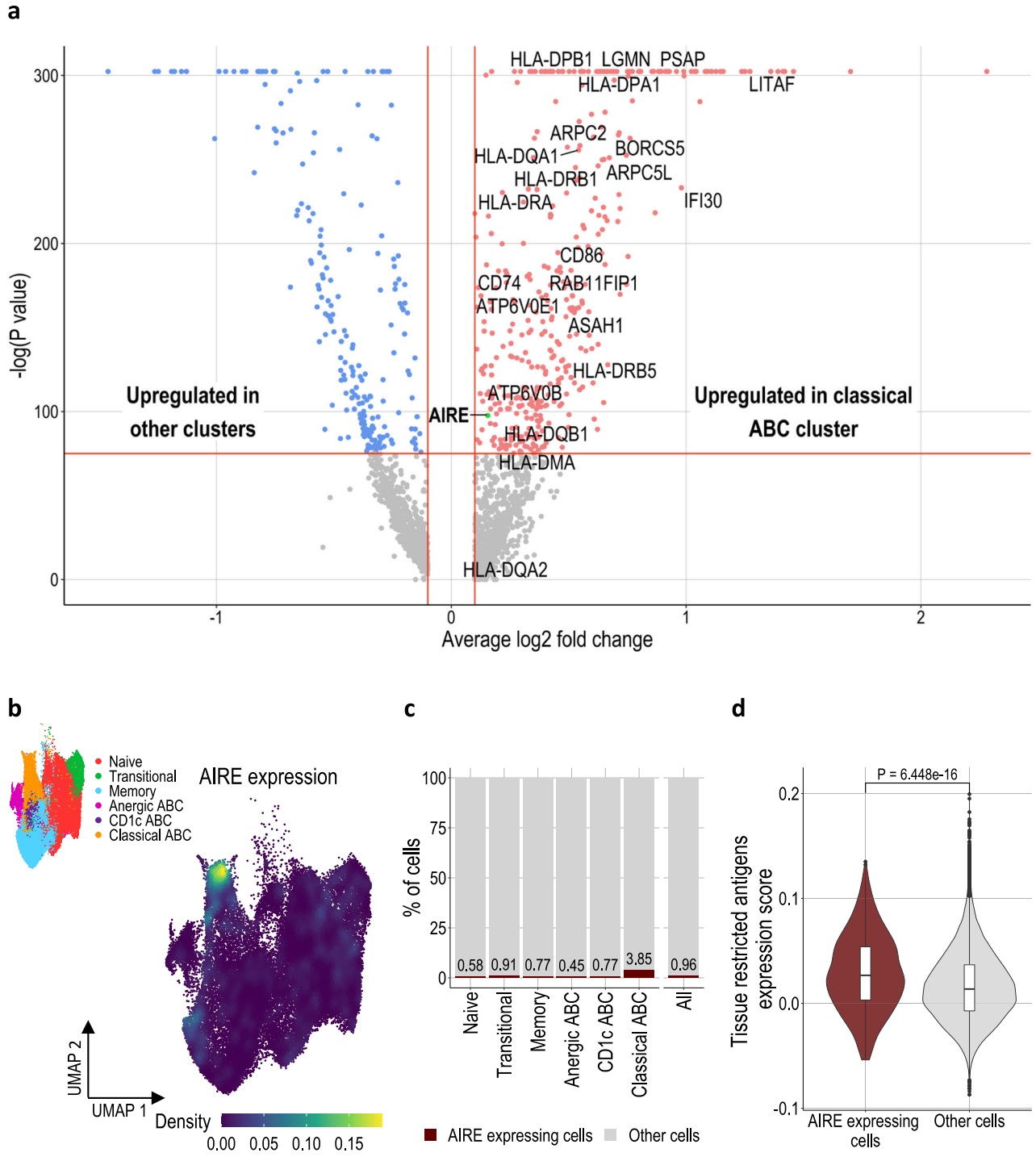

**Fig. 2 | Classical ABC expression of AIRE and AIRE target genes. a** Volcano plot displaying the differentially expressed genes in ABCs compared to other B cell clusters. Upregulated genes associated with antigen uptake, processing and MHC class II presentation are labelled. AIRE gene is coloured in green. Statistical testing via two-tailed Wilcoxon rank sum test with Bonferroni correction. **b** UMAP of all B cells coloured by kernel density estimation of AIRE expression level across all identified B cell subsets. UMAP showing the different B cell clusters inset. **c** Percentage of cells which are AIRE⁺ in each cluster. **d** Violin plot comparing expression score of tissue restricted antigens (gene set from Yamano et al.) in AIRE expressing cells (n = 503) and all other cells (n = 51899). In the boxplot, the centre, lower and upper bounds of the box correspond to the median, first quartile and third quartile respectively. The upper(lower) whisker extends from the box to the largest(smallest) value no further than 1.5 * IQR from the box upper(lower) bound (where IQR is the inter-quartile range, or distance between the first and third quartiles). Statistical testing via one-tailed Wilcoxon rank sum test.

Supplementary Table 3). All participants were invited for blood sampling prior to their second vaccine dose (day 0), then at early (day 8), mid (day 21) and late (day 105) time points after this dose (Fig. 3a). We assessed serum samples 24 h after the second dose of BNT162b2 and

did not observe any difference in the levels of IL-1β, IL-12 and IFN-α between patients and controls at this time point, suggesting a similar innate immune response to the vaccine (Supplementary Fig. 6). Classical ABC frequencies prior to vaccination (day 0) were determined by

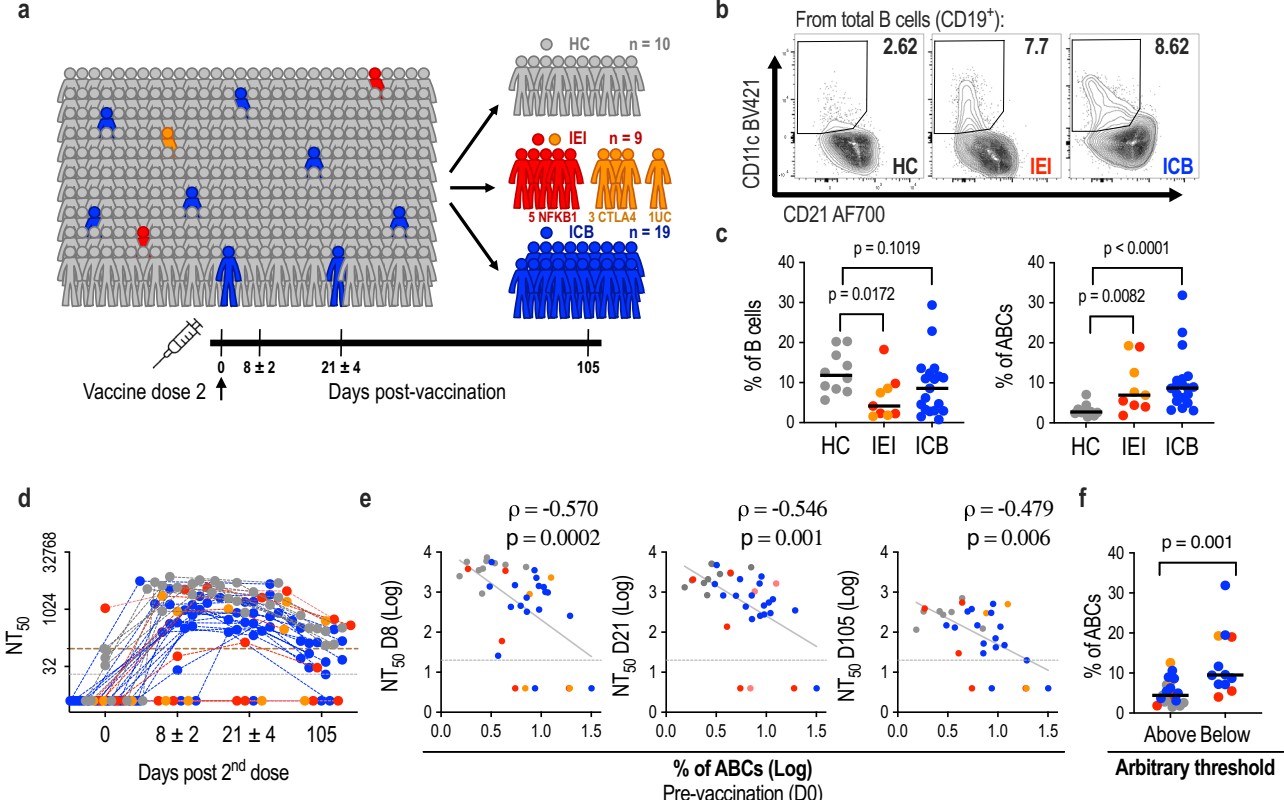

**Fig. 3 | Correlation between frequency of ABCs and neutralising antibody response to COVID-19 vaccination. a** Cohort details. Samples were collected at days 0, 8, 21 and 105 after the 2nd dose of BNT162b2 vaccine (healthy controls (HC), grey; patients with rare inborn errors of immunity (IEI): NFKB1 red, CTLA-4 and unclassified orange; patients treated with ICB, blue). **b** Representative FACS contour plots of CD21lo CD11c+ ABCs in CD19 B cells. **c** Frequencies of B cells within total lymphocytes and frequencies of ABCs within total B cells, at day 0. HC = 10, IEI = 9, ICB = 19 biologically independent samples. Differences between groups were determined using two-tailed non-parametric Mann–Whitney tests. **d** Neutralising antibody titres at 50% inhibition (NT$_{50}$) against wildtype SARS-CoV-2 at indicated

timepoints after 2nd vaccine dose. The limit of detection of the assay is indicated (grey dotted line at NT$_{50}$ = 20), and an arbitrary threshold at the highest NT$_{50}$ from the HC group at day 0 (brown dashed line). **e** Correlations between frequencies of ABCs amongst B cells at day 0 and NT$_{50}$s at days 8, 21 and 105. Two-tailed Spearman's rank correlation coefficients (rho) and p values are shown, together with indicative linear regression lines. **f** Frequencies of ABCs amongst CD19+ B cells for individuals above and below the arbitrary threshold indicated in panel **d** at day 105. Above = 19, Below = 12 biologically independent samples. Differences between groups were determined using two-tailed non-parametric Mann–Whitney tests. **c**–**f** each point represents one individual.

multicolour flow cytometry staining for CD11c and CD21 (Supplementary Fig. 7a) and found to be elevated in patients compared with healthy controls (Fig. 3b, c).

Serum reactivity against SARS-CoV-2 spike (S), receptor-binding domain (RBD) and nucleocapsid (NCP) antigens was evaluated at day 21 after the second dose in the diagnostic immunology laboratory. The majority of patients (18/19 ICB, 5/8 IEI) generated IgG antibodies against S and RBD antigens (encoded by BNT162b2) at levels similar to the healthy control group (Supplementary Fig. 7b). These responses were unlikely to reflect previous natural infection with SARS-CoV-2, as patients did not have evidence of NCP reactivity (not encoded by BNT162b2) (Supplementary Fig. 7b). The quality of the SARS-CoV-2-specific antibody response can be determined by measuring the capacity of serum to neutralise authentic SARS-CoV-2[48]. Overall, most donors displayed a rapid increase in neutralising antibody titres, peaking early (day 8) after their second vaccine dose (Fig. 3d and Supplementary Fig. 7c). By contrast, 5 patients (4 IEI and 1 ICB) failed to develop detectable neutralising capacity at any time point, despite having detectable anti-S antibodies. These patients are at greater risk of infection with SARS-CoV-2 and may be susceptible to prolonged or refractory COVID-19[49,50].

Critically, the frequency of pre-vaccine ABCs was inversely correlated with neutralising antibody titre at all timepoints (Fig. 3e), suggesting that ABCs predict both peak neutralising capacity after vaccination and the longevity of the neutralising antibody response.

Indeed, at day 105, 4/8 patients with IEI and 8/16 patients treated with ICB had neutralising antibody titres which had fallen below the highest titre observed at day 0 in healthy controls, compared with 0/7 HCs (Fig. 3d). Individuals with neutralising capacity below this threshold showed higher frequencies of ABCs than the individuals above it (Fig. 3f).

A similar negative correlation between ABC frequency and neutralising capacity was still observed when we subdivided the cohort according to sex (Supplementary Fig. 8a), when the analysis was restricted to those participants with ABC frequencies falling within the range observed in healthy controls (Supplementary Fig. 8b), and when the analysis was restricted to those donors with B cell frequencies falling within the range of healthy controls (Supplementary Fig. 8c). Premature expansion of ABCs is therefore seen both in patients with IEI and patients with cancer treated with ICB, and (regardless of aetiology) correlates with a reduced ability to generate and maintain neutralising antibody responses to COVID-19 vaccination.

## Premature expansion of ABCs is associated with lower levels of antigen-specific memory B cells
In addition to circulating antibodies derived from plasma cells, long-lived immunological memory can persist in expanded clones of antigen-specific memory B cells, both arms of B cell responses can better counteract the same pathogen during subsequent encounters. We therefore assessed the frequency and differentiation of circulating

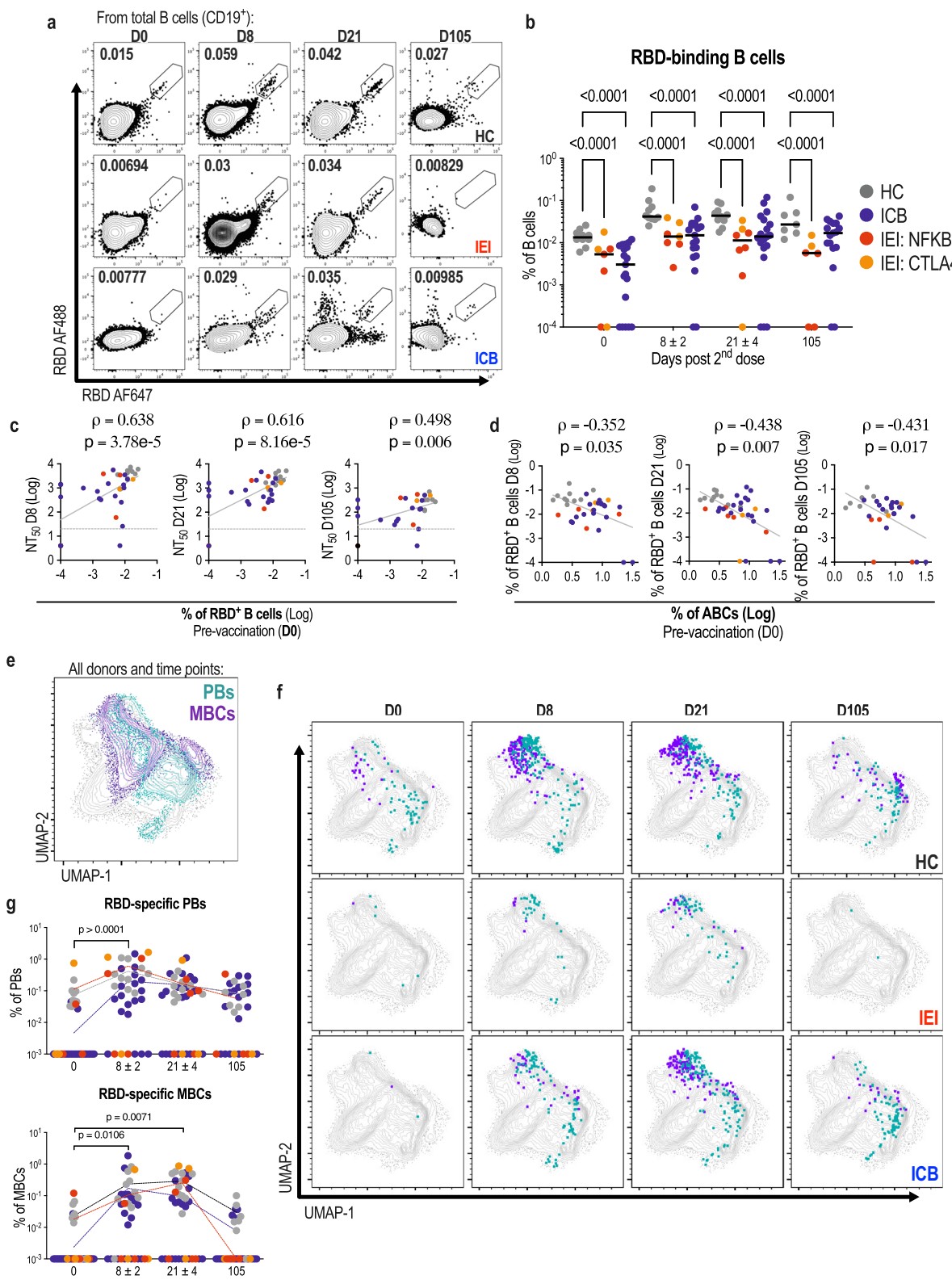

SARS-CoV-2 RBD-specific B cells elicited after the second dose of vaccine using multicolour flow cytometry (Supplementary Fig. 9a). Strikingly, RBD-binding B cells were significantly less frequent in the different patient groups than in healthy controls (Fig. 4a, b). mRNA-LNP immunisation is able to induce antigen-specific ABCs[24], often termed atypical memory B cells. We therefore queried whether a high pre-vaccine ABC frequency is associated with induction of vaccine-antigen specific ABCs. Conversely, we found that the frequency of RBD-specific ABCs was reduced in all patient groups (Supplementary Fig. 9b). The fraction of RBD-binding B cells amongst all CD19+ B cells correlated consistently with titres of neutralising antibodies (Supplementary Fig. 9c). In addition, RBD-specific B cell frequency immediately prior to the second vaccine dose correlated with neutralising capacity at days 8 (peak response), 21 and 105 (Fig. 4c).

**Fig. 4 | Correlation between frequency of RBD-specific B cells and neutralising antibody response to COVID-19 vaccination. a** Representative flow cytometry plots displaying RBD-specific B cell populations amongst total CD19$^+$ B cells for different study groups 0, 8, 21 and 105 days (D0, D8, D21 and D105) after the second dose of BNT162b2 vaccine. **b** Frequencies of RBD-specific B cells amongst all CD19$^+$ B cells in HC, IEI and ICB patients at days 0, 8, 21 and 105. Each dot represents a single individual (healthy controls (HC, D0 = 10, D8 = 10, D21 = 10, D105 = 7), grey; patients with rare inborn errors of immunity (IEI, D0 = 7, D8 = 7, D21 = 8, D105 = 6): (NFKB1) red, (CTLA-4 and unclassified) orange; patients treated with ICB (ICB, D0 = 19, D8 = 19, D21 = 19, D105 = 17), blue). Two-way ANOVA with Tukey's multiple comparisons test for statistical analysis. **c** Correlations between frequencies of RBD-specific B cells amongst all CD19$^+$ B cells at day 0 and NT$_{50}$s at days 8, 21 and 105. Two-tailed Spearman's rank correlation coefficients (rho) and $p$ values are

shown, together with indicative linear regression lines. **d** Correlations between frequencies of ABCs amongst B cells at day 0 and frequencies of RBD-specific B cells at days 8, 21 and 105. Two-tailed Spearman's rank correlation coefficients (rho) and p values are shown, together with indicative linear regression lines. **e** UMAP projection of total B cells from all donors at all time points displaying memory B cell (MBCs, purple) and plasmablast (PBs, green) populations. **f** RBD-specific B cells with MBC (purple) or PB (green) phenotype are shown at days 0, 8, 21 and 105 (columns) for HC (top), IEI (middle) and ICB (bottom) groups. **g** Kinetics of RBD-binding cell frequencies amongst MBCs or PBs. Statistical significance between groups was determined using ordinary one-way ANOVA. Samples with no detectable levels of RBD-specific cells are plotted at an arbitrary value of $10^{-4}$ in **b** and $10^{-3}$ in **g**.

This is consistent with a causal relationship between the level of pre-existing antigen-specific B cells, and the magnitude and longevity of the neutralising antibody response upon re-encounter with the antigen. Furthermore, similar to neutralising capacity, the levels of RBD-specific B cells at later timepoints could be predicted by the frequency of ABCs at day 0 (Fig. 4d). Individuals with the highest frequencies of ABCs showed reduced RBD-specific B cell differentiation (Supplementary Fig. 9c, d).

To further analyse the cellular phenotype of rare RBD-binding B cells in a comprehensive, unbiased way, we performed unsupervised clustering and UMAP visualisation of flow cytometry data from all patients and healthy controls at all timepoints after vaccination (Fig. 4e). RBD-binding B cells from both patients and healthy controls were observed within clusters of memory B cells and plasmablasts (Fig. 4f), suggesting a quantitative reduction rather than obvious qualitative difference in the humoral response. Across all participants, the proportion of RBD-specific cells expressing plasmablast markers peaked at day 8, whilst those expressing memory B cell makers peaked a day 21 (Fig. 4g). Differentiation of antibody-producing cells therefore occurs contemporaneously with peak neutralising capacity (day 8). Since increased antibody production cannot be driven by the proliferation of terminally differentiated (non-dividing) plasma cells, a sizeable contribution to the neutralising antibody response after the second dose of vaccine is likely the result of this memory B cell proliferation and differentiation.

Amongst patients treated with ICB, we observed similar RBD-specific B cell frequencies and neutralising responses when stratified by steroidal treatment, dual/single checkpoint blockade therapy and cancer stage (III/IV) (Supplementary Fig. 10a, b). Conversely, we noted a reduction in both RBD specific B cell frequency and the neutralising antibody response with increasing age (Supplementary Fig. 10c). In theory, several factors (such as comorbidity) could impede the vaccination responses of older individuals with cancer, independent of ABC frequency. We therefore specifically examined the associations between ABC frequency, age and diminished humoral response amongst individuals under the age of 60. In this subset, RBD-specific B cell frequency and neutralising capacity were still negatively correlated with ABC frequency, but not with age (Supplementary Fig. 10d). These results support a primary association between pre-vaccine ABC frequency and impairment of the humoral response.

### T cell assessment following second dose of mRNA BNT162b2 vaccination

Humoral vaccine responses require B and T lymphocyte interaction, we next assessed T cells. Circulating T follicular helper cell (Tfh) frequencies were actually increased in patients with IEI and unchanged in patients treated with ICB (Supplementary Fig. 11a, b). Similarly, activated (OX40$^+$CD137$^+$) CD4$^+$ T cell frequency was increased in patients with IEI before vaccination and this was maintained at day 21. No significant difference was observed in CD8$^+$ T cell activation markers, nor

in patients treated with ICB (Supplementary Fig. 11c). To assess antigen-specific responses, we next enumerated Spike-specific T cells using IFN-γ ELISpot (Supplementary Fig. 11d). No significant differences were observed between healthy controls and patient groups. Finally, we assessed the correlation between neutralising capacity and pre-vaccination levels of circulating Tfhs but found no significant association (Supplementary Fig. 11e). Taken together, these results suggest that the association between increased ABC frequency and diminished humoral vaccine response is independent of T cell function.

### Assessment of ABCs for immune complex binding and cytokine secretion

Preclinical evidence has suggested that ABCs might impede humoral vaccine responses by diminishing affinity maturation[51]. Affinity maturation is a process requiring iterative selection of proliferating B cell clones within the germinal centre, and this takes time. However, we found substantial neutralising capacity within 8 days of vaccination, suggesting a limited requirement for substantial further affinity maturation. Furthermore, the negative correlation between ABC frequency and neutralising capacity was greatest at early time points after repeat immunisation. We therefore investigated other mechanisms by which ABCs could limit humoral vaccine responses. Further scRNAseq analysis indicated that ABCs, particularly classical ABCs, expressed high levels of the inhibitory Fc gamma receptor IIB (*FcγRIIB* or *CD32B*) (Fig. 5a). Expression of other Fc receptors was not observed (Supplementary Fig. 12a) consistent with the literature suggesting this is the only Fc receptor expressed on B lymphocytes[52]. To test this functionally, we generated immune complexes by incubation of fluorescent rabbit anti-human IgG with polyclonal human IgG at a ratio of 1:2. This resulted in complexes three times the molecular weight of rabbit IgG, consistent with a trimolecular antibody stoichiometry (Supplementary Fig. 12b). The binding of these complexes to B cells was Fc-dependent and significantly increased in ABCs from all patient cohorts relative to other B cell subsets (Fig. 5b, d). This positions ABCs as the B cell subset best placed to clear immune complexes directed against vaccine antigen, potentially reducing the longevity of antigen availability. Furthermore, the differentiation of RBD-specific ABCs may be inhibited by signalling through the FcγRIIB limiting their contribution to memory B cell expansion or antibody secreting cell differentiation. Production of cytokines has been suggested as one mechanism by which ABCs limit B cell function[53,54] and in preclinical models of *NFKB1* deficiency B cell production of IL-6 contributes to disease pathogenesis[55]. We therefore evaluated cytokine production capacity by in-vitro stimulation of B cells subsets with PMA/ionomycin and evaluated intracellular cytokines production by FACS (Fig. 5c and Supplementary Fig. 12c). This showed that although ABCs from all patient groups can produce IL-6 and TNF (Fig. 5e), the frequency of these was not higher than in other B cell subsets. Taken together within our transcriptional assessment, these results indicate ABCs from different patient groups are functionally similar and present additional

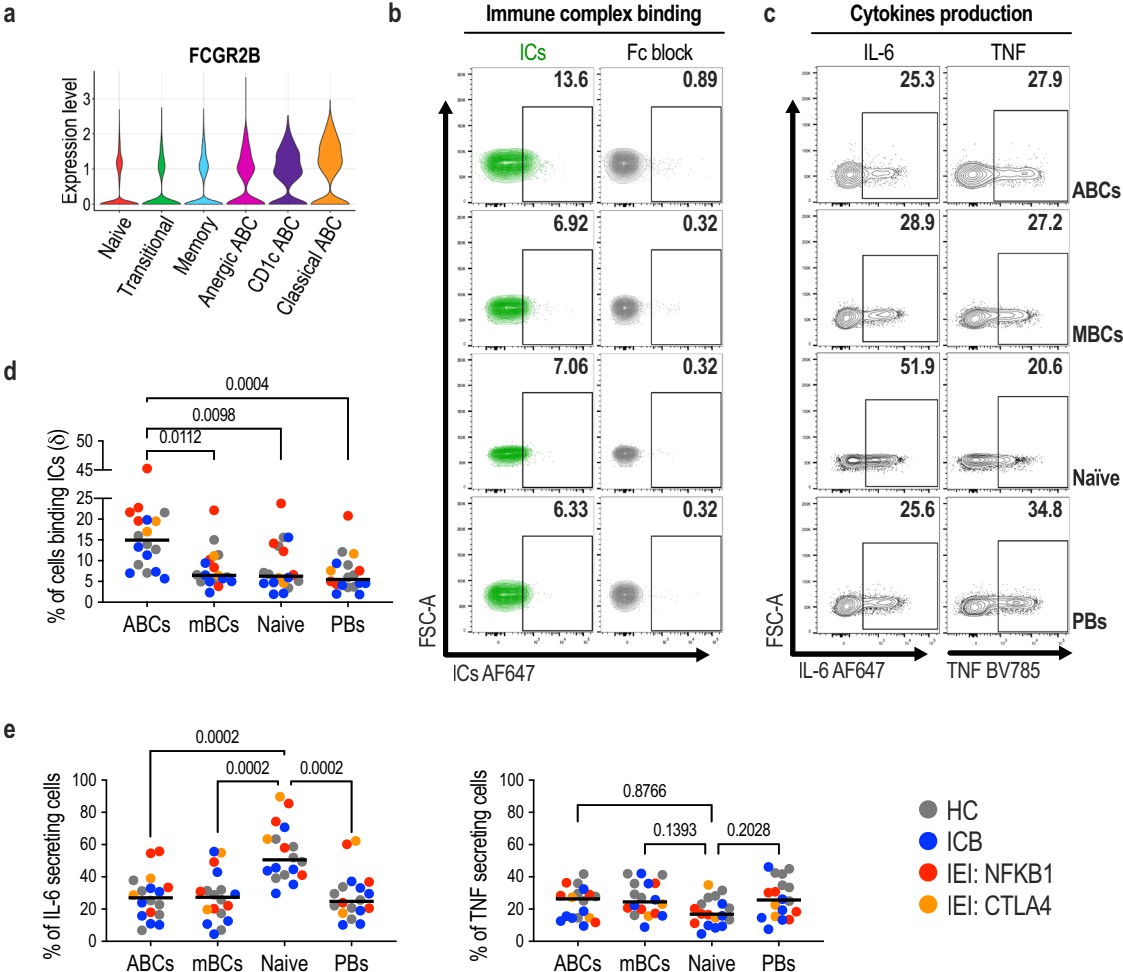

**Fig. 5 | ABCs express higher levels of Fc γ receptor IIB and bind higher proportions of immune complexes than other B cells subsets. a** Violin plots displaying the expression levels of *FCGR2B* on different B cell subsets. **b** Representative flow cytometry contour plots displaying immune complex binding (contour plots in green) with or without a previous incubation with Fc block by age-associated B cells (ABCs), memory B cells (MBCs), naïve B cells and plasmablasts (PBs). **c** Representative flow cytometry contour plots displaying cytokines production (IL-6 and TNF, PMA/ionomycin stimulation for 5 h, 37 °C, brefeldin A) by ABCs, MBCs, naïve B cells and PBs. **d** Frequencies of ICs binding cells in different B cell subsets. **e** Frequencies of cells with IL-6- and TNF-secreting capacity in different B cell subsets. Where specified, statistical significance between groups was determined using one-way ANOVA. **d**, **e** HC = 6, IEI = 6, ICB = 6 biologically independent samples.

mechanisms by which elevated ABCs may diminish humoral vaccine responses.

## Discussion

ABC differentiation is a physiological component of the humoral response to intracellular pathogens and the vaccines that protect against them[13]. Exogenous antigen and Type I inflammatory signals trigger ABC differentiation and class switching necessary for antibody driven cell mediated cytotoxicity. Conversely, in the absence of exogenous antigen recognition and BCR signalling, TLR7 stimulation can lead to expansion of ABCs with self-reactivity[16]. In that setting ABCs may be directed against endogenous antigens, including self-antigens[16,56]. These distinct causes for ABC expansion emphasise that in any single older patient the cause for expansion (to which polygenic risk alleles, ageing, therapy, infection exposure vaccines and obesity can all contribute) is unknown. However, in young patients with ultra-rare monogenic inherited disease, expansion is caused by the specific gene disrupted. Our analysis of 4 different patient groups: healthy controls, patients with cancer treated with ICB, patients with autoimmune SLE, and patients with two distinct IEI (CTLA-4 haploinsufficiency and NFKB1 haploinsufficiency) demonstrates a homogeneity of the ABC differentiation states irrespective of the specific cause.

Therefore, the pathological consequence of expanded ABCs is likely related to their increased frequency rather than an inherent difference in these cells from patients with distinct diseases. In this study we demonstrate that one of these consequences is a limitation of the humoral immune response to SARS-CoV-2 vaccination. ABCs arising from either B cell intrinsic or extrinsic reasons in preclinical models have been shown to reduce vaccine responses by impairing affinity maturation of the germinal centre response[51]. However, this mechanism alone may not account for a neutralisation difference already apparent 8 days after repeat immunisation. Our finding that ABCs express high levels of the inhibitory FcγRIIB receptor and are the B cell subset best able to bind immune complexes indicates additional mechanisms by which ABCs may limit vaccine responses.

There are several limitations of this study. First, the number of people studied was modest, limiting the power to detect small differences in some comparisons and precluding a formal multivariate analysis. Second, many characteristics of ABCs may contribute to an impaired humoral vaccine response, including (but not limited to) reduced affinity maturation, signalling through inhibitory FcγRIIB and clearance of antigen. We have not quantified the extent of affinity maturation in this study, nor yet performed the detailed preclinical experiments necessary to determine the relative contribution of these

mechanisms. Third, absolute numbers of ABCs were not quantified in this study, it is not possible to ascertain whether the number of ABCs or the skewing of the B cell repertoire towards this differentiation state, leads to the impairment in vaccine responses. Finally, we cannot be certain that a breach of self-tolerance has triggered the ABC expansion in patients with *NFKB1* and *CTLA4* haploinsufficiency, since this expansion could also be reactive to intracellular pathogens (to which these patients may be more susceptible).

Despite these limitations our finding that patients with increased ABC frequency have a reduced B cell vaccine response, leading to reduced neutralising capacity and reduced memory B cell formation is of immediate clinical relevance. Overall, these results place ABC frequency as a predictive biomarker for reduced vaccine protection which could guide booster vaccination schedules for patients at risk of breakthrough infection.

# Methods

## Patient recruitment and ethics

Enrolment of patients into this study was based on the deficiency of particular genes of interest (IEI: *CTLA4*, *LRBA*, *NFKB1*, *NFKB2*) or clinical diagnostics (ICB, $n = 19$), whilst healthy controls for this study were enroled based on their clinical healthy status ($n = 10$). Sample-size calculation was limited due to IEI patients in the study ($n = 9$) reflect a rare population of individuals within the general population that we were able to recruit (Supplementary Table 3).

The research was conducted in accordance with the principles of Good Clinical Practice and following approved protocols of the NIHR National Bioresource. Samples were collected with the written informed consent of all study participants under the NIHR National BioResource - Research Tissue Bank (NBR-RTB) ethics (REC:17/EE/0025) or under the Barts and the London Immunology Registry (REC: 11/LO/1689). Clinical data were collected by Clinical Immunology Consultants at Cambridge University Hospital and Bart's Health via the Electronic Healthcare Record (Epic), or direct patient contact.

The patients and healthy controls were consented under the East of England Cambridge South national research ethics committee (REC) reference 13/EE/0325 or Barts and the London Immunology Registry (REC: 11/LO/1689).

## Blood donation and separation

Patients and controls gave voluntary blood samples prior to second dose and at early (day 8), mid (day 21) and late (day 105) timepoints following their second BNT162b2 vaccine. Where sample size of individual assays was not the same as the total sample size, this was because there was no sample collection at those specific time points. Samples may have also been excluded if there was serological evidence of prior exposure to SARS-CoV-2. Peripheral blood samples were acquired in either lithium heparin or serum separating tubes. Peripheral blood mononuclear cells (PBMCs) were isolated by a density gradient centrifugation protocol and stored at −80 °C for up to a week before being transferred to liquid nitrogen until use. Whilst serum tubes were centrifuged to separate serum from cell pellet before being aliquoted and stored at −80 °C until use.

## Single cell library preparation and sequencing

Frozen PBMCs were thawed and B cell population was enriched using magnetic separation (Miltenyi 130-101-638). Samples were next labelled with oligonucleotide-tagged antibodies against CD11c, CD21, CD27 and CD85j (TotalSeq, BioLegend). After labelled, cell counts were adjusted to 1035 cells per microlitre before being loaded onto a Chromium Next GEM chip K (10X Genomics, 2000182) for the subsequent 5′ Gene Expression and V(D)J libraries' construction according to Single cell 5′ v2 protocol (CG000330 RevA, 10X Genomics). The library was quality controlled by Agilent 4150 TapeStation and quantified by RT-PCR using the KAPA library quantification kit for Illumina

platforms (Roche, KK4824). Samples were sequenced on an Illumina NovaSeq with a sequencing depth of at least 50, 000 reads per cell.

## Software versions

Single cell transcriptome and VDJ data was analysed using Cell Ranger software version 6.1.1 and 7.0.1 respectively, citeseq version 1.5.0, R version 4.0.3 and R packages (Seurat 4.1.0, SeuratObject 4.0.4, Scuttle 1.0.4, rstatix 0.7.0, tidyverse 1.3.0 and dplyr 1.0.8) and python packages (dandelion 0.3.0, pandas 1.1.3, numpy 1.19.2, networkx 2.5, scipy 1.5.2, blast 2.11.0, igblast 1.17.1, anndata 0.7.6, scanpy 1.7.2, changeo 1.0.2 and airr 1.3.1). Flow cytometry data and antibody titres were analysed using FlowJo 10.7.1 and GraphPad Prism 9.1.2 (225), respectively. FCS files were analysed using FlowJo 10.7.2. Additionally, a uniform manifold approximation and projection (UMAP) algorithm for dimensional reduction was performed on a concatenated FCS file comprising events from the CD19$^+$ gate, utilising the UMAP FlowJo plugin (v3.1). XShift (v1.4.1) and ClusterExplorer (1.5.15) plugins were used for unsupervised cluster generation and exploration, respectively. Figures were produced using ggplot2 3.3.5, gridExtra 2.3, ComplexHeatmap 2.6.2, Nebulosa 1.0.2, RColorBrewer 1.1-2, ggrepel 0.9.1, ggpubr 0.4.0, scales 1.1.1, showtext 0.9-5 in R, GraphPad Prism 9.1.2 (225) and FlowJo 10.7.1.

## Preprocessing of scRNA-seq transcriptome data

Raw FASTQ files of the gene expression library were analysed using 10x Genomics Cell Ranger software v.6.1.1[57] and aligned to the GRCh38 genome provided by Cell Ranger. All Ig V, D, J and constant genes as well as TCR genes were deleted from the dataset so that downstream analysis is not affected by highly variable clonotype genes. For the quantification of antibody derived tags (ADT) and oligo hashtags, citeseq pipeline v.1.5.0[58] was used. The generated hashtag count matrices as well as raw gene expression count matrices were then used as inputs for the Seurat package v.4.1.0[59] in R v.4.0.3. Seurat objects were created from the corresponding transcript count matrices. ADT assays were added using CreateAssayObject function of Seurat to include expression levels of surface proteins. Samples were then demultiplexed using HTODemux function. Doublets and cells with no assigned hashtag were removed from objects for further analysis. Scuttle package v.1.0.4[60] was used for quality control. 3 x Median absolute deviation (MAD) was considered as the threshold for quality control; number of features and number of read counts were filtered from both sides, whereas percentage of mitochondrial genes was filtered from upper side. The samples were log-normalised for further differential expression analysis. Before integrating all objects, they were normalised based on regularised negative binomial regression using the SCTransform function of the Seurat package, regressing out for cell cycle scores, number of counts and percentage of mitochondrial genes. The objects were then integrated using the integration protocol of the Seurat package to perform batch effect correction. The objects used for integration include: four groups of healthy controls each having two participants (total of 26900 cells in the final dataset), one group of IEI patients (5345 cells in the final dataset) and two groups of ICB patients (total of 9751 cells in the final dataset) each having four participants, and SLE patients with 3 participants (10406 cells in the final dataset).

## SLE data

Data of SLE patients from Bhamidipati et al.[33] were downloaded from GSE163121, reprocessed and integrated with other objects using the same procedure as described above.

## Clustering of data

Principal component analysis (PCA) was performed on the integrated object to reduce the dimensionality of the dataset. First, clustering was done using 30 principal components (PCs) and resolution parameter

of 0.1. Clusters with a high expression of non-B cell markers were removed from the dataset. After deletion of all non-B cells, the dataset consists of 52402 cells. PCs were calculated in the new dataset and unsupervised clustering was performed using 20 PCs and resolution parameter of 1.4. Cluster markers were calculated with the default parameters of the FindMarkers function. Clusters with similar markers were manually merged. Final clusters were labelled based on the differentially expressed genes in each cluster.

### Gene ontology enrichment analysis
Gene ontology[61,62] enrichment analysis was performed via PANTHER[63] from http://www.geneontology.org using the list of significantly upregulated genes from classical ABC cluster (Average log2 fold change > 0.25 and percent expressed in each group > 0.25).

### Processing of scRNA-seq VDJ data
Raw FASTQ files of the VDJ library were analysed using the 10x Genomics Cell Ranger software v7.0.1 and aligned to the VDJ GRCh38 genome provided by Cell Ranger. The Dandelion package was then used for analysis of clonal expansion in different clusters[38,39]. Using dandelion V/D/J genes were re-annotated, heavy V gene alleles and constant region calls were re-assigned and cells with poor quality contigs were filtered[64,65]. Clones were defined based on following criterion:
  i. usage of the same V and J genes
  ii. identical CDR3 sequence length
  iii. minimum of 85% sequence similarity between CDR3 sequences based on hamming distance

Dandelion was then used for generating a tree-like network for each clone based on the similarities of the full VDJ contig sequences. Vertex/node Gini index was calculated as a measure of clonal expansion in different clusters[37]. For calculating Gini index, dandelion merges identical BCRs into one node in the BCR network and the number of merged nodes is counted. The degree index from dandelion was also calculated as a measure of how many cells are connected to an individual cell in the clonal network. Both indices can vary between 0 and 1.

### Geneset scores
A list of 74 Aire-induced genes were obtained from Yamano et al.[41], after converting mouse gene IDs to human gene IDs (Supplementary Table 2). For calculating gene set score for each cell, AddModuleScore function from Seurat package was used. Briefly, average expression of genes in each cell is calculated and is subtracted by the aggregated expression of control gene sets. For selecting control gene sets, genes are first binned. Then, for each gene, control gene sets are randomly chosen from the same expression bin as that gene, so that they have similar expression patterns to Aire target genes. To explore the expression pattern of these genes in different tissues, normalised transcript per million (nTPM) values were extracted in different immune cells and different tissues from the human protein atlas website (https://v22.proteinatlas.org/download/rna_immune_cell.tsv.zip and https://v22.proteinatlas.org/download/rna_tissue_consensus.tsv.zip, respectively). In addition to two groups of B lymphocytes, those tissues and cells that have maximum expression of at least one of the genes from our list were kept.

### Kernel density estimation of gene expression
Nebulosa package v 1.0.2[66] was used for calculating and plotting gene-weighted density estimation of AIRE expression.

### SARS-CoV-2 serology
A Luminex bead-based immunoassay was used to quantify specific antibodies to full-length trimeric spike (S), spike receptor binding domain (RBD) and nucleocapsid (NCP) of SARS-CoV-2 as previously described[67,68]. Briefly, a multiplex assay was established by covalently coupling recombinant SARS-CoV-2 proteins to distinct carboxylated bead sets (Luminex, Netherlands). The S protein used here was the S-R/PP described in Xiong et al.[67], and the RBD protein was described by Stadlbauer et al.[69]. The NCP protein used is a truncated construct of the SARS-CoV-2 NCP protein comprising residues 48–365 (both ordered domains with the native linker) with an N terminal uncleavable hex-ahistidine tag. NCP was expressed in *E. Coli* using autoinducing media for 7 h at 37 °C and purified using immobilised metal affinity chromatography (IMAC), size exclusion and heparin chromatography. The S-, RBD- and NCP-coupled bead sets were incubated with patient sera at 3 dilutions (1/100, 1/1000, 1/10000) for 1 h in 96-well filter plates (MultiScreen HTS; Millipore) at room temperature in the dark on a horizontal shaker. After washes, beads were incubated for 30 min with a PE-labelled anti-human IgG-Fc antibody (Leinco/Biotrend), washed as described above, and resuspended in 100 µl PBS/Tween. Antibody-specific binding was interpreted using Exponent Software V31 software on the Luminex analyzer (Luminex/R&D Systems) and reported as mean fluorescence intensity (MFI). The diagnostic thresholds used adhered to UK national guidelines.

### Neutralising antibodies to SARS-CoV-2
The SARS-CoV-2 used in this study was a wildtype (lineage B) virus (SARS-CoV-2/human/Liverpool/REMRQ0001/2020), a kind gift from Ian Goodfellow (University of Cambridge), isolated early in the COVID-19 pandemic by Lance Turtle (University of Liverpool) and David Matthews and Andrew Davidson (University of Bristol)[70–72] from a patient on the Diamond Princess cruise ship. Luminescent HEK293T-ACE2-30F-PLP2 reporter cells (clone B7) expressing ACE2 and SARS-CoV-2 Papain-like protease-activatable circularly permuted firefly luciferase (FFluc) are available from the National Institute for Biological Standards and Control (NIBSC, www.nibsc.org, catalogue number 101062)[48]. Sera were heat-inactivated at 56 °C for 30 min before use, and neutralising antibody titres at 50% inhibition ($NT_{50s}$) measured as previously described[48,73].

In brief, luminescent HEK293T-ACE2-30F-PLP2 reporter cells were seeded in flat-bottomed 96-well plates. The next day, SARS-CoV-2 viral stock (MOI = 0.01) was pre-incubated with a 3-fold dilution series of each serum for 2 h at 37 °C, then added to the cells. 16 h post-infection, cells were lysed in Bright-Glo Luciferase Buffer (Promega) diluted 1:1 with PBS and 1% NP-40, and FFluc activity measured by luminometry. Experiments were conducted in duplicate.

To obtain $NT_{50s}$, titration curves were plotted as FFluc *vs* log (serum dilution), then analysed by non-linear regression using the Sigmoidal, 4PL, X is log(concentration) function in GraphPad Prism. $NT_{50s}$ were reported when (1) at least 50% inhibition was observed at the lowest serum dilution tested (1:20), and (2) a sigmoidal curve with a good fit was generated. For purposes of visualisation and ranking, samples for which visual inspection of the titration curve indicated inhibition at low dilutions, but which did not meet criteria (1) and (2) above, were assigned an arbitrary $NT_{50}$ of 4.

World Health Organisation International Standard 20/136 (WHO IS 20/136) has an NT50 of 1967 against wildtype SARS-CoV-2 when measured in this assay[74]. This standard comprises pooled convalescent plasma obtained from 11 individuals which, when reconstituted, is assigned an arbitrary neutralising capacity of 1000 IU/ml against early 2020 SARS-CoV-2 isolates[75]. $NT_{50s}$ from this study may therefore be converted to IU/ml using a calibration factor of 1000/1967 (0.51), with a limit of quantitation of 10.2 IU/ml (corresponding to an $NT_{50}$ of 20).

### Flow cytometric analysis
Frozen PBMCs were thawed and stained with specific antibody cocktails (Supplementary Table 4). All samples were acquired on the BD LSRFortessa using FACSDIVA software (BD-Biosciences). FCS files were exported and analysed using FlowJo v10.7.2 (BD-Biosciences) software. Additionally, a uniform manifold approximation and projection (UMAP)

algorithm for dimensional reduction was performed on a concatenated FCS file comprising events from the CD19+ gate from HC, IEI and ICB samples analysed, utilising the UMAP FlowJo plugin (v3.1). The composite samples were gated as indicated on Supplementary Figures for RBD+ B cell identification and then overlaid by study group in the UMAP generated, for additional visualisation. XShift (v1.4.1) and ClusterExplorer (1.5.15) plugins were used for unsupervised cluster generation and exploration, respectively. Class-switched memory B cells and plasmablasts/plasma cells were selected based on the expression of CD19+IgD-IgM-CD38-CD27+ and CD19+IgD-IgM-CD38+/high CD27+/high, respectively.

### RBD tetramer production
SARS-CoV-2 RBD was expressed with N-terminal fusion of His-Zbasic-TEV module (8×His tag, Zbasic domain[76] and TEV protease cleavage site) and C-terminal Avi-tag in BL21(DE3) cells (Novagen) cultured in 2xYT media at 37 °C. RBD containing inclusion bodies were isolated and solubilised. The denatured protein was first purified by immobilised metal affinity chromatography using PureCube Ni-NTA resin, eluted and refolding was allowed to proceed for 72 h at 4 °C. N-terminal fusion tag was cleaved off by NHis-TEVpro (produced in-house). Biotinylation of the RBD protein was carried out using 5 mM MgCl₂, 2 mM ATP, 150 µM biotin and 50 µg/ml of biotin ligase BirA-CHis (produced in-house). The excess of biotinylated RBD protein was then incubated with 0.5 mg/ml of fluorescently labelled Streptavidin (BioLegend) for 1 h at room temperature and, finally, RBD-Streptavidin complexes were separated by size exclusion chromatography using a Superdex 200 Increase 10/300 GL column (Cytiva). The RBD-Streptavidin complexes were then analysed by SDS-PAGE.

### Immune complex binding assay
Immune complexes were generated by incubation of 2 µg of human immunoglobulins (Biolegend 422302) with 1 µg of Alexa Fluor 647 AffiniPure Rabbit Anti-Human IgG (Jackson ImmunoResearch 309-605-006) for 1 h at room temperature and diluted 1 in 10 before cell binding. Immune complex size was analysed by means of hydrodynamic radius ($R_h$) measured by Microfluidic Diffusional Sizing (MDS) on a Fluidity One-W (Fluidic Analytics) instrument. Thawed PBMCs were or not incubated with Fc block for 30 min before interaction with ICs for 10 min at room temperature. After immune complex binding, PBMCs were fixed with PFA at 1% for 30 min and then washed. After fixation, PBMCs were stained with a B cell antibody cocktail panel for surface labelling. All samples were acquired on the BD LSRFortessa using FACSDIVA software (BD-Biosciences). FCS files were exported and analysed using FlowJo v10.7.2 (BD-Biosciences) software.

### In vitro cytokine analysis
Frozen PBMCs were thawed and activated with phorbol 12-myristate 13-acetate (PMA; 50 ng/ml, Sigma-Aldrich) and Ionomycin (1 µM, Sigma-Aldrich) for 5 h at 37 °C. Inhibitor of intracellular protein transport, Brefeldin A (eBioscience), was added to cells at the start of stimulation. Cells incubated with vehicle (DMSO) and Brefeldin A were used as unstimulated controls. Cytokines production was analysed by flow cytometry. Briefly, cells were incubated with LIVE/DEAD fixable Aqua dead cell stain (Thermo Fisher Scientific) and Fc block, followed by cell-surface marker staining. Cells were then fixed and permeabilised using fixation/permeabilization buffer (BD Biosciences). Antibodies to stain intracellular cytokines were added and incubated for 30 min at room temperature. All samples were acquired on the BD LSRFortessa using FACSDIVA software (BD-Biosciences). FCS files were exported and analysed using FlowJo v10.7.2 (BD-Biosciences) software.

### T cell cytokine production by ELISpot
Samples were assessed at day 105 after 2nd dose BNT162b2 immunisation. Frozen PBMCs were thawed and rested for 2–3 h in RPMI media

supplemented with 10% (v/v) Human AB Serum (Sigma) and 1% (v/v) Penicillin/Streptomycin (Sigma) at 37 °C. Plates precoated (Mabtech ELISpot Plus: Human IFN-γ (ALP), 3420-4APT-10) with capture antibody (Mabtech, mAb 1-D1K) were washed three times with PBS and then blocked with supplemented RPMI media at 37 °C for 1–2 h. Overlapping peptide pools (18-mers with 10 amino acid overlap. Mimotopes) representing the spike (S), Membrane (M) or nucleocapsid (N) SARS-CoV-2 proteins were added to 200,000 PBMCs/well at a final concentration of 2 µg/ml. Concanavalin A (Sigma) was used as positive control. DMSO (Sigma) was used as the negative control at the equivalent concentration to the peptides. Plates were incubated at 37 °C, for 18 h. Wells were washed with PBS 0.05% (v/v) Tween (Sigma) seven times before incubation for 2 h at room temperature with the ELISpot PLUS kit biotinylated detection antibody (clone 7-B6-1) at 1 µg/ml. Wells were washed and then incubated with the ELISpot PLUS kit streptavidin-ALP at 1 µg/ml for 1 h at room temperature. Wells were washed and colour development was carried out using 1-step NBT/BCIP Substrate Solution, for 5 min at room temperature. Colour development was stopped by washing the plates with cold tap water. Plates were left to air dry for 48 h and then scanned and analysed using the AID iSpot Spectrum ELISpot reader (software version 7.0, Autoimmune Diagnostika GmbH, Germany). The average spot count of the control wells was subtracted from the test wells for each sample to quantify the antigen-specific responses. Results are expressed as spot forming units (SFU) per 10⁶ PBMCs.

### Statistical analysis
Statistical analysis for FACS and serology data was performed using GraphPad Prism v9.1.1 (GraphPad Prism Software Inc). Two-tailed non-parametric Mann–Whitney tests were used to determine differences between groups. For multiple comparisons ANOVA with Tukey's multiple comparisons test was used. Associations were calculated using two-tailed Spearman's rank correlations and results are shown with linear trend levels. For the single cell data, statistical significance for paired comparisons were performed using Wilcoxon rank sum test in R.

### Reporting summary
Further information on research design is available in the Nature Portfolio Reporting Summary linked to this article.

## Data availability
The raw and processed sequencing data generated in this study have been deposited in the Gene Expression Omnibus (GEO) database under accession code GSE207475. The SLE data used in this study are available in the GEO database under accession code GSE163121. Gene expression data in different immune cells and different tissues are extracted from the human protein atlas website (https://v22.proteinatlas.org/download/rna_immune_cell.tsv.zip and https://v22.proteinatlas.org/download/rna_tissue_consensus.tsv.zip, respectively). GRCh38 genome and GRCh38 VDJ genome are downloaded from https://cf.10xgenomics.com/supp/cell-exp/refdata-gex-GRCh38-2020-A.tar.gz and https://cf.10xgenomics.com/supp/cell-vdj/refdata-cellranger-vdj-GRCh38-alts-ensembl-7.0.0.tar.gz, respectively. All other data are available in the article and its Supplementary files or from the corresponding author upon request. Source data are provided with this paper.

## Code availability
All scripts used for processing the sequencing data and generating figures in this work are deposited in Zenodo (https://doi.org/10.5281/zenodo.7806635)[77].

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

## Acknowledgements

This work was funded by the UK Medical Research Council (project number MC_UU_00025/12), the Medical Research Foundation (MRF-057-0002-RG-THAV-C0798) and The Evelyn Trust (grant number 20/40) to J.E.D.T. N.J.M. was supported by the MRC (TSF ref. MR/T032413/1), NHSBT (grant ref. WPA15-02) and Addenbrooke's Charitable Trust (grant ref. 900239). M.A.C. was supported by the Medical Research Council (project number MC_UU_00025/10). K.R.P. was supported by the Medical Research Council (project number MC_UU_00025/11). K.F. held an MRC studentship with support from the Cambridge European Trust and St. John's College. K.W. has received funding by the Deutsche Forschungsgemeinschaft (WA 1597/6-1 and WA 1597/7-1). K.W. and B.K. received support by the German Federal Ministry of Education and Research (BMBF) through a grant to the German genetic multi-organ Auto-Immunity Network (GAIN), grant code 01GM2206A. F.H. is an ERC Advanced Investigator (695669). We thank Carola G. Vinuesa for helpful discussion. The authors also thank the Flow Cytometry Facilities at the MRC-Toxicology Unit, University of Cambridge; Katarzyna Kania from CRUK-CI-Genomics, Cambridge UK for advice on single cell RNA sequencing experiments; and Rosalind Kieran from the Department of Oncology, Cambridge University NHS Hospitals Foundation Trust, Cambridge UK, for contributions for patient recruitment and data collection.

## Author contributions

J.C.Y.-P., Z.H., E.C.H., P.P.G., N.B.-C., R.H. cell isolation, designed and performed experiments, analysis e interpretation of results. A.L., K.F. RBD tetramer production. M.R., R.B., M.A., E.S.-R., M.O.R., H.R., L.H.B., L.K., A.C., G.S.F.A., X.-M.S., T.E.M., L.P.G.P. cell isolation and technical support. A.C.-N., D.F., L.C.-G., A.E., C.S., A.B., S.J., N.K., S.G., S.S., E.S. recruitment of participants of the study and sample collection. B.K., M.C., M.M., A.E.W., K.R.P., K.W., M.S.B., F.H., M.H., R.D. data discussion. C.P., S.L., N.J.M., J.E.D.T. provided oversight on recruitment of participants, sample collection and experiments. J.C.Y.-P., Z.H., J.E.D.T. prepared the figures and wrote the manuscript. J.C.Y.-P., J.E.D.T. designed and supervised the study. J.C.Y.-P., Z.H., E.C.H., P.P.G., N.B.-C., R.H. contributed equally. C.P., S.L., N.J.M., J.E.D.T. contributed equally. All authors approved the final version of the manuscript.

## Competing interests

The authors declare no competing interests.

## Additional information

Juan Carlos Yam-Puc [1,15] ✉, Zhaleh Hosseini[1,15], Emily C. Horner [1,15], Pehuén Pereyra Gerber[2,3,15], Nonantzin Beristain-Covarrubias[1,15], Robert Hughes[1,15], Aleksei Lulla[4], Maria Rust[1], Rebecca Boston[1], Magda Ali [1], Katrin Fischer [4], Edward Simmons-Rosello[1], Martin O'Reilly [1], Harry Robson[1], Lucy H. Booth[1], Lakmini Kahanawita[1], Andrea Correa-Noguera[5], David Favara[5], Lourdes Ceron-Gutierrez[6], Baerbel Keller[7,8], Andrew Craxton[1], Georgina S. F. Anderson[1], Xiao-Ming Sun[1], Anne Elmer[9], Caroline Saunders[9], Areti Bermperi[9], Sherly Jose[9], Nathalie Kingston[10], Thomas E. Mulroney[1], Lucia P. G. Piñon[1], CITIID-NIHR COVID–19 BioResource Collaboration*, Michael A. Chapman[1], Sofia Grigoriadou[11], Marion MacFarlane[1], Anne E. Willis[1], Kiran R. Patil[1], Sarah Spencer[1], Emily Staples[1,6], Klaus Warnatz [7,8,12], Matthew S. Buckland [11,13], Florian Hollfelder [4], Marko Hyvönen [4], Rainer Döffinger [6], Christine Parkinson[5,16], Sara Lear[6,16], Nicholas J. Matheson [2,3,14,16] & James E. D. Thaventhiran [1,6,16] ✉

[1]Medical Research Council Toxicology Unit, School of Biological Sciences, University of Cambridge, Cambridge, UK. [2]Cambridge Institute of Therapeutic Immunology and Infectious Disease (CITIID), University of Cambridge, Cambridge, UK. [3]Department of Medicine, University of Cambridge, Cambridge, UK. [4]Department of Biochemistry, University of Cambridge, Cambridge, UK. [5]Department of Oncology, Cambridge University NHS Hospitals Foundation Trust, Cambridge, UK. [6]Department of Clinical Immunology, Cambridge University NHS Hospitals Foundation Trust, Cambridge, UK. [7]Department of Rheumatology and Clinical Immunology, Medical Center - University of Freiburg, Faculty of Medicine, University of Freiburg, Freiburg, Germany. [8]Center for Chronic Immunodeficiency (CCI), Medical Center - University of Freiburg, Faculty of Medicine, University of Freiburg, Freiburg, Germany. [9]NIHR Cambridge Clinical Research Facility, Cambridge, UK. [10]NIHR BioResource, Cambridge University Hospitals NHS Foundation Trust, Cambridge, UK. [11]Department of Clinical Immunology, Barts Health, London, UK. [12]Department of Immunology, University Hospital Zurich, Zurich, Switzerland. [13]UCL GOSH Institute of Child Health Division of Infection and Immunity, Section of Cellular and Molecular Immunology, London, UK. [14]NHS Blood and Transplant, Cambridge, UK. [15]These authors contributed equally: Juan Carlos Yam-Puc, Zhaleh Hosseini, Emily C. Horner, Pehuén Pereyra Gerber, Nonantzin Beristain-Covarrubias, Robert Hughes. [16]These authors jointly supervised this work: Christine Parkinson, Sara Lear, Nicholas J. Matheson, James E. D. Thaventhiran. *A list of authors and their affiliations appears at the end of the paper. ✉e-mail: jcy28@cam.ac.uk; jedt2@cam.ac.uk

## CITIID-NIHR COVID−19 BioResource Collaboration

Juan Carlos Yam-Puc [1,15] ✉, Zhaleh Hosseini[1,15], Emily C. Horner [1,15], Nonantzin Beristain-Covarrubias[1,15], Robert Hughes[1,15], Maria Rust[1], Rebecca Boston[1], Lucy H. Booth[1], Anne Elmer[9], Caroline Saunders[9], Areti Bermperi[9], Sherly Jose[9], Nathalie Kingston[10], Thomas E. Mulroney[1], Sarah Spencer[1], Nicholas J. Matheson [2,3,14,16] & James E. D. Thaventhiran [1,6,16] ✉

A full list of members and their affiliations appears in the Supplementary Information.

