## [Peer Review File · Nature Communications]

Age-Associated B cells predict impaired humoral immunity after COVID-19 vaccination in patients receiving immune checkpoint blockadeREVIEWER COMMENTS

Reviewer #1 (COVID-19 and immune deficiency) (Remarks to the Author):

Yam-Puc et al. This is a technically well done study that focuses on the impact of circulating levels of human aging associated B cells (ABC) on IgG antibody responses to second dose of COVID-19 Spike second mRNA dosing. ABC of different subsets are dissected and enumerated by multiparameter flow cytometry and CITEseq with surface phenotyping antibodies. The central finding is that no matter the cause of increased ABC levels, whether it is due to genotypic deficiency (NFKB1 and CTLA4 deficiency) or aging, or checkpoint inhibitor (i.e. PD-1 inhibitor) treatment, if increased ABC expansions are present before a second vaccine dose, this has a powerful adverse impact on patient vaccination responses for the generation of neutralizing antibodies. The comparisons were to Spike or RBD binding in a high sensitivity bead-based Luminex assay, which were less affected by the levels of ABCs.

The observations are powerful and the analyses of several B cell subsets is convincing. The problem with the report is that the mechanism(s) responsible for the observed functional defect in generation of post -vaccination neutralizing antibody responses remain completely unexplained. This paper is more of a beginning than a complete story.

One can speculate that high affinity neutralizing antibodies are predominantly the product of T cell dependent germinal center responses, which take time and antigen specific T Follicular helper cells (TFH). The authors do not comment on what is known about ABC and viral antigen-specific responses. Yet it has been reported that ABCs are TLR7-mediated responders and may not be vaccine antigen driven.

To evaluate the possibility that the defect is truly in T cell dependent GC responses, one wonders whether ABC were found to bind tetramers of Spike or RBD antigens? What is the frequency? Alternatively, are the ABC at best low affinity antibody bearing and secreting upon stimulation? The single cell (10X Genomics) studies could have yielded paired VH:VL usage and evidence of clonal expansions in ABC vs memory B cells. Hence antigen-specific recombinant antibodies could be reconstituted and tested for kinetics/affinity/avidity of Spike or RBD domain binding.

While lymph node or tonsillar biopsies would be required to characterize the human response to this clinically important vaccine, quantitation of peripheral TFH and in vitro T cell Spike peptide rechallenge studies, or TRUculture studies, might provide more clarity as to whether high ABC levels is associated with impaired antigen specific T cell responses.

Minor point

Is it known why NFK1 deficiency (described as intrinsic) causes ABC expansions and CTLA4 (described as extrinsic) deficiency causes ABC expansions?

Reviewer #2 (Age-associated B cell) (Remarks to the Author):

In this manuscript submitted by Thaventhiran et al. the relationship between the course of generating age-associated B cells (ABC) is investigated by comparing the outcome of treatment with immune checkpoint blockade (ICB) or other conditions (inherited or SLE as prototypic autoimmune disease), and correlating these data with the occurrence of ABCs and antigen-specific memory B cells and antibody responses following vaccination. Their results demonstrate that ABCs in individuals with the above conditions (healthy or disease-affected) have shared features in the ABC by scRNAseq which, however, can be clustered in various subsets, potentially correlating with diverse immune functions. In this regard the expression of AIRE in one subset (classical ABCs) seems noteworthy. Importantly the frequency of ABCs inversely correlated with the production of neutralizing antibodies against SARS-Covid-2 RBD, and also the emergence of antigen-specific memory B cells. These observations can further augment our understanding the impact of ICB on the clinical course and efficacy of Covid vaccination in these patients.

While the manuscript is generally well written (with appropriate and current methodology and

statistical analyses), several additions need to be considered for its acceptance.

(1) The description of ICB patients is lacking (what kind of cancers at what stage the patients had; what type of ICBs/administration stage they received; age and biological gender composition, etc.); therefore it is difficult to appreciate their sample homogeneity.

(2) The authors report that the ABC frequency is higher in patients with SLE, similarly to a higher frequency found in females amongst healthy controls. SLE is known to have a female dominance, thus raising whether the association is linked to the existence of autoimmune disease itself, or rather it reflects its higher occurrence amongst females. Have the patients within the SLE group also been stratified according to their biological gender?

(3) Throughout the manuscript the numeric parameters for B cells/ABCs and their subsets are expressed as frequency; given that the frequency of B cells is higher, while the frequency of ABCs is lower in healthy individuals [Fig. 3c] (and the opposite is found in both immune deficient and ICB patients), yet can the absolute number of ABCs/unit of blood be similar in all three groups?

(4) In Fig 4a/b, the authors claim that "Strikingly, RBD-binding B cells were always significantly less frequent in patients than in healthy controls"; one panel of the graphs shows actually a higher frequency of Ag-specific B cells in ICB patient on D21 [0,042 vs. 0,054] (in panel a), and there is no indication of significance (in panel b) supporting the claim. Furthermore, the fold-increase of these cells in both the IEI and ICB groups actually seems higher than in the healthy controls - please comment.

(5) Were there any attempts to perform in vitro stimulation tests on B cells, to compare various groups for investigating the degree of autonomous B-cell intrinsic preference for ABC formation? This would be important to support the claim in conclusion on the necessity for repeated Covid vaccination in patients with ICB, lest the immunization elicits a disproportionately larger number of ABCs with dysfunctional features, that would compromise the efficiency of repeated vaccination.

Reviewer #3 (B cell biology) (Remarks to the Author):

The manuscript by Yam-Puc et al. compared the transcriptional profile of age-associated B cells (ABCs) across cohorts of healthy controls, cancer patients receiving immune checkpoint blockade (ICB) therapy and patients with NFkB or CTLA-4 haploinsufficiency (IEI). They found that ABCs were transcriptionally similar across the different cohorts but elevated in ICB and IEI patients. They found that the ABCs expressed AIRE and displayed an increased profile of tissue-restricted antigens, which is typically restricted to the thymus. The increased frequency of ABCs correlated with decreased response to Covid19 vaccinations, including decreased maintenance of neutralizing antibodies.

The observation that ABCs display a similar transcriptional profile between the healthy controls and ICB and IEI patients is interesting. However, the subsequent figures in the manuscript rely completely on correlations, drawing conclusions of the role of ABCs in vaccine induced Ab production that is not supported by any functional assays. There are likely a number of other factors that can impact maintenance of the antibody response in these patients as they display either genetically, or therapy, induced alterations in their broader immune response, which will include CD4 T cells that help in antibody production.

Minor comments:

The abstract sounds like 2 different studies rather than looking at ICB on B cells and confirming phenotypes in vaccinated patients.

Figure 2d, a better definition of what the tissue restricted antigen score is. At least a supplemental table and expression pattern.

Reviewer #4 (Immune cell biology, transcriptome analyses) (Remarks to the Author):

In this manuscript, Yam-Puc et. al. studies the effect of age-associated B cells and the response to covid vaccination. Their study groups include healthy control, cancer patients treated with immune checkpoint blockade, patients with NFKB1 and CTLA4 haploinsufficiency, and SLE. They identified 1) common transcriptional profiles for ABCs in the study groups, 2) distinct ABCs clusters between groups, 3) classical ABCs express AIRE, and 4) ABCs frequency predicts neutralizing Ab responses to COVID vaccination. Based on these observations, the authors conclude that ABC frequency might be a predictive biomarker for reduced vaccine protection. The data is interesting, but mostly descriptive and don't provide any mechanistic explanation for the observed results.

Comment #1: Also, the sample size is small. Most of the observed findings need to be confirmed in a larger N. For example, in figure 3c, about 40% of IEI and 50% of ICB subjects has similar % of B cells compared to HC, but the authors shoed some significant differences between HC and IEI. Similar observation is also true for % of ABCs graph. Only few patients have higher frequency of ABCs. Thus, more N is needed to confirm these results.

Comment #2: the identification of ABCs is not optimal, as ABCs are commonly identified by the expression of T-bet, CD11c, CD11b, and lack of CD21. It is not clear why the authors only used CD11c and CD21 as markers to define ABCs?

Comment #3: In figure 2a, it's not clear why the authors only focused on AIRE gene induction? There were many other genes that were upregulated in ABCs. The authors need to at least discuss the relevance of some of other differentially expressed/upregulated genes.

Comment #4: In figure 4a-b, the author showed some minor differences in ABCs between HC and patients, but it is necessary to also show functional differences such as production of IFN γ , TNF, IL-17, IL-10, or IL-4 cytokines or other readouts. Is there any functional differences in cytokine profile between vaccinated and unvaccinated subject? And does the disease state affects this response?

Responses to reviewers' comments:

Reviewer #1 (Remarks to the Author)

This is a technically well done study that focuses on the impact of circulating levels of human aging associated B cells (ABC) on IgG antibody responses to second dose of COVID-19 Spike second mRNA dosing. ABC of different subsets are dissected and enumerated by multiparameter flow cytometry and CITEseq with surface phenotyping antibodies. The central finding is that no matter the cause of increased ABC levels, whether it is due to genotypic deficiency (NFKB1 and CTLA4 deficiency) or aging, or checkpoint inhibitor (i.e. PD-1 inhibitor) treatment, if increased ABC expansions are present before a second vaccine dose, this has a powerful adverse impact on patient vaccination responses for the generation of neutralizing antibodies. The comparisons were to Spike or RBD binding in a high sensitivity bead-based Luminex assay, which were less affected by the levels of ABCs.

We thank the reviewer for their kind words, and agree with their central point. Titres of neutralising antibodies are correlated with protection against infection with SARS-CoV-2¹ and may be dissociated from levels of binding antibodies to spike and/or the spike RBD². Numerous studies have identified biomarkers associated with progression and/or severity of COVID-19³⁻⁵. Nonetheless, to our knowledge, expansion of ABCs is the first immune biomarker to be associated with impaired antibody responses to SARS-CoV-2 vaccination, across a range of different conditions.

The observations are powerful and the analyses of several B cell subsets is convincing. The problem with the report is that the mechanism(s) responsible for the observed functional defect in generation of post -vaccination neutralizing antibody responses remain completely unexplained. This paper is more of a beginning than a complete story.

We thank the reviewer for suggesting we address this point. By conducting a further analysis and new experiments we have identified that ABCs express high levels of *FCGR2B* and this is associated with an increased ability to bind immune complexes. This is now displayed in Fig. 5.

Correspondingly, we have added the following text to the results on page 10:

“Assessment of ABCs for immune complex binding and cytokine secretion

Preclinical evidence has suggested that ABCs might impede humoral vaccine responses by diminishing affinity maturation⁵¹. Affinity maturation is a process requiring iterative selection of proliferating B cell clones within the germinal centre, and this takes time. However, we found substantial neutralising capacity within 8 days of vaccination, suggesting a limited requirement for substantial further affinity maturation. Furthermore, the negative correlation between ABC frequency and neutralising capacity was greatest at early time points after repeat immunisation. We therefore investigated other mechanisms by which ABCs could limit humoral vaccine responses. Further scRNAseq analysis indicated that ABCs, particularly classical ABCs, expressed high levels of the inhibitory Fc gamma receptor IIB (Fc γ RIIB or CD32B) (Fig. 5a). Expression of other Fc receptors was not observed (Supplementary Fig. 12a), consistent with the literature suggesting this is the only Fc receptor expressed on B lymphocytes⁵². To test this functionally, we generated immune complexes by incubation of fluorescent rabbit anti-human IgG with polyclonal human IgG at a ratio of 1:2. This resulted in complexes three times the molecular weight of rabbit IgG, consistent with a trimolecular antibody stoichiometry (Supplementary Fig. 12b). The binding of these complexes to B cells was Fc-dependent, and significantly increased in ABCs from all patient cohorts relative to other B cell subsets

(Fig. 5b and 5d). This positions ABCs as the B cell subset best placed to clear immune complexes directed against vaccine antigen, potentially reducing the longevity of antigen availability. Furthermore, the differentiation of RBD-specific ABCs may be inhibited by signalling through the FcγRIIB, limiting their contribution to memory B cell expansion or antibody secreting cell differentiation. Production of cytokines has been suggested as one mechanism by which ABCs limit B cell function^{53,54} and in preclinical models of NFKB1 deficiency B cell production of IL-6 contributes to disease pathogenesis⁵⁵. We therefore evaluated cytokine production capacity by in-vitro stimulation of B cells subsets with PMA/ionomycin and evaluated intracellular cytokines production by FACS (Fig. 5c and Supplementary Fig. 12c). This showed that although ABCs from all patient groups can produce IL-6 and TNF-α (Fig. 5e), the frequency of these was not higher than in other B cell subsets. Taken together within our transcriptional assessment, these results indicate ABCs from different patient groups are functionally similar and present additional mechanisms by which elevated ABCs may diminish humoral vaccine responses.”

In addition, we have introduced the following statement of study limitations to page 11 of the discussion:

“There are several limitations of this study. First, the number of people studied was modest, limiting the power to detect small differences in some comparisons, and precluding a formal multivariate analysis. Second, many characteristics of ABCs may contribute to an impaired humoral vaccine response, including (but not limited to) reduced affinity maturation, signalling through inhibitory FcγRIIB and clearance of antigen. We have not quantified the extent of affinity maturation in this study, nor yet performed the detailed preclinical experiments necessary to determine the relative contribution of these mechanisms. Third, absolute numbers of ABCs were not quantified in this study, it is not possible to ascertain whether the number of ABCs or the skewing of the B cell repertoire towards this differentiation state, leads to the impairment in vaccine responses. Finally, we cannot be certain whether a breach of self-tolerance has triggered the ABC expansion in patients with NFKB1 and CTLA4 haploinsufficiency, since this expansion could also be reactive to intracellular pathogens (to which these patients may be more susceptible).”

One can speculate that high affinity neutralizing antibodies are predominantly the product of T cell dependent germinal center responses, which take time and antigen specific T Follicular helper cells (TFH). The authors do not comment on what is known about ABC and viral antigen-specific responses. Yet it has been reported that ABCs are TLR7-mediated responders and may not be vaccine antigen driven.

We thank for the reviewer for their suggestion. We agree, and have added this point to the discussion on page 10-11:

“ABC differentiation is a physiological component of the humoral response to intracellular pathogens, and the vaccines that protect against them¹³. Exogenous antigen and Type I inflammatory signals trigger ABC differentiation and class switching necessary for antibody driven cell mediated cytotoxicity. Conversely, in the absence of exogenous antigen recognition and BCR signalling, TLR7 stimulation can lead to expansion of ABCs with self-reactivity¹⁶. In that setting, ABCs may be directed against endogenous antigens, including self-antigens^{16,56}.”

To evaluate the possibility that the defect is truly in T cell dependent GC responses, one wonders whether ABC were found to bind tetramers of Spike or RBD antigens? What is the frequency?

We thank for the reviewer for their suggestion. We have now evaluated the proportion RBD-specific ABCs, and found it to be reduced in patients, compared with healthy controls, at all timepoints (D0, D8, D21 and D105 after the second vaccine dose). This is in keeping with ABCs in patients being predominantly directed towards other antigens including (potentially) self-antigens. This also suggests that having high levels of ABCs pre-vaccination does not lead to increased antigen-specific ABC induction after antigen challenge. These data are displayed in Sup. Fig. 9b:

Coresspondingly, we have added the following text to the results on page 8:

“mRNA-LNP immunisation is able to induce antigen-specific ABCs²⁴, often termed atypical memory B cells. We therefore queried whether a high pre-vaccine ABC frequency is associated with induction of vaccine-antigen specific ABCs. Conversely, we found that the frequency of RBD-specific ABCs was reduced in all patients groups (**Supplementary Fig. 9b**).”

Alternatively, are the ABC at best low affinity antibody bearing and secreting upon stimulation? The single cell (10X Genomics) studies could have yielded paired VH:VL usage and evidence of clonal expansions in ABC vs memory B cells.

We thank the reviewer for this suggestion. We have now analysed the VDJ library using the 10x Genomics Cell Ranger software. Details of this analysis are explained in the Material and Methods. This analysis does indeed show evidence of clonal expansions in classical ABCs, and that these clones are related. These data are displayed as Sup. Fig. 3:

Correspondingly, we have added the following text to the results on page 6:

“To further analyse these different clusters, we used single cell BCR sequencing data to calculate the vertex/node Gini index as a measure of clonal expansion³⁷, and the degree index from dandelion^{38,39} as a measure of how cells are clonally related. The classical ABC cluster had the highest Gini index, and the second highest degree index, suggesting that more clones of classical ABCs are expanded, and that these clones are related (**Supplementary Fig. 3**).”

Hence antigen-specific recombinant antibodies could be reconstituted and tested for kinetics/affinity/avidity of Spike or RBD domain binding.

We thank the reviewer for their suggestion, and agree that decreased affinity maturation (as suggested by Zhang et al.⁶) could contribute to the poor humoral vaccine response in people with elevated ABCs. Whilst directly testing this hypothesis is beyond the scope of this paper, we now discuss this possibility in the results on page 10:

“Preclinical evidence has suggested that ABCs might impede humoral vaccine responses by diminishing affinity maturation⁵¹. Affinity maturation is a process requiring iterative selection of proliferating B cell clones within the germinal centre, and this takes time. However, we found substantial neutralising capacity within 8 days of vaccination, suggesting a limited requirement for substantial further affinity maturation. Furthermore, the negative correlation between ABC frequency and neutralising capacity was greatest at early time points after repeat immunisation.”

In addition, we have highlighted it in the new statement of study limitations on page 11:

“Second, many characteristics of ABCs may contribute to an impaired humoral vaccine response, including (but not limited to) reduced affinity maturation, signalling through inhibitory FcγRIIB and clearance of antigen. We have not quantified the extent of affinity maturation in this study, nor yet performed the detailed preclinical experiments necessary to determine the relative contribution of these mechanisms.”

While lymph node or tonsillar biopsies would be required to characterize the human response to this clinically important vaccine, quantitation of peripheral TFH and in vitro T cell Spike peptide rechallenge studies, or TRUculture studies, might provide more clarity as to whether high ABC levels is associated with impaired antigen specific T cell responses.

We thank the reviewer for this suggestion. Accordingly, we have assessed circulating TFH frequency, T cell activation, antigen specific ELISpot responses and the correlation between circulating TFH numbers and neutralising capacity. These data are displayed in Sup. Fig. 11:

Correspondingly, we have added the following text to the results on page 9:

“Humoral vaccine responses require B and T lymphocyte interaction, we next assessed T cells. Circulating T follicular helper cell (Tfh) frequencies were actually increased in patients with IEI and unchanged in patients treated with ICB (Supplementary Fig. 11a-b). Similarly, activated (OX40+CD137+) CD4 T cell frequency was increased in patients with IEI before vaccination, and this was maintained at day 21. No significant difference was observed in CD8+ T cell activation markers, nor in patients treated with ICB (Supplementary Fig. 11c). To assess antigen-specific responses, we next enumerated Spike-specific T cells using IFN- γ ELISpot (Supplementary Fig. 11d). No significant differences were observed between healthy controls and patient groups. Finally, we assessed the correlation between neutralising capacity and pre-vaccination levels of circulating Tfh, but found no significant association. Taken together, these results suggest that the association between increased ABC frequency and diminished humoral vaccine response is independent of T cell function.”

Minor point

Is it known why NFK1 deficiency (described as intrinsic) causes ABC expansions and CTLA4 (described as extrinsic) deficiency causes ABC expansions?

We thank the reviewer for highlighting this question. The antigenic reactivity of the expanded ABCs in these conditions is unknown. We have now noted this in the new statement of study limitations on page 11.

“Finally, we cannot be certain whether a breach of self-tolerance has triggered the ABC expansion in patients with NFKB1 and CTLA4 haploinsufficiency, since this expansion could also be reactive to intracellular pathogens (to which these patients may be more susceptible).”

Reviewer #2 (Remarks to the Author)

In this manuscript submitted by Thaventhiran et al. the relationship between the course of generating age-associated B cells (ABC) is investigated by comparing the outcome of treatment with immune checkpoint blockade (ICB) or other conditions (inherited or SLE as prototypic autoimmune disease), and correlating these data with the occurrence of ABCs and antigen-specific memory B cells and antibody responses following vaccination. Their results demonstrate that ABCs in individuals with the above conditions (healthy or disease-affected) have shared features in the ABC by scRNAseq which, however, can be clustered in various subsets, potentially correlating with diverse immune functions. In this regard the expression of AIRE in one subset (classical ABCs) seems noteworthy. Importantly the frequency of ABCs inversely correlated with the production of neutralizing antibodies against SARS-Covid-2 RBD, and also the emergence of antigen-specific memory B cells. These observations can further augment our understanding the impact of ICB on the clinical course and efficacy of Covid vaccination in these patients. While the manuscript is generally well written (with appropriate and current methodology and statistical analyses), several additions need to be considered for its acceptance.

(1) The description of ICB patients is lacking (what kind of cancers at what stage the patients had; what type of ICBs/administration stage they received; age and biological gender composition, etc.); therefore it is difficult to appreciate their sample homogeneity.

We thank the reviewer for this suggestion, and have therefore now re-analysed the humoral vaccine response in patients with cancer subsetted by a number of relevant variables. In brief, whilst the majority of these variables were not associated with a significant difference in either NT50 or RBD-binding B cell frequency, we did observe a correlation between age and humoral response. Prior studies have suggested that this association is particularly evident in those of advanced age⁷. We therefore specifically re-analysed our data from younger study participants (age <60). Amongst those individuals, the inverse correlation between ABC frequency and humoral response was maintained, whilst that between age and humoral response was not. This suggests that the primary association is with ABC frequency, rather than age per se. These data are displayed in Sup. Fig. 10:

Correspondingly, we have added the following text to the results on page 9:

“Amongst patients treated with ICB, we observed similar RBD-specific B cell frequencies and neutralising antibody responses when stratified by steroidal treatment, dual/single checkpoint blockade therapy and cancer stage (III/IV) (Supplementary Fig. 10a-b). Conversely, we noted a reduction in both RBD-specific B cell frequency and the neutralising antibody response with increasing

age (Supplementary Fig. 10c). In theory, several factors (such as comorbidity) could impede the vaccination responses of older individuals with cancer, independent of ABC frequency. We therefore specifically examined the associations between ABC frequency, age and diminished humoral response amongst individuals under the age of 60. In this subset, RBD-specific B cell frequency and neutralising capacity were still negatively correlated with ABC frequency, but not with age (Supplementary Fig. 10d). These results support a primary association between pre-vaccine ABC frequency and impairment of the humoral response.”

In addition, we have summarised the characteristics of these patients in Sup. Table 5:

Table S5. Characteristics of Inborn Errors of Immunity (IEI) patients and Immune Checkpoint blockade treated (ICB) patients

Condition	Age	Sex	Current Treatment	Most recent IgG (6-16 g/L)	NT ₅₀ (D21)	% ABCs
Healthy Controls						
Healthy Controls (n=10)	31-57	F (5), M(5)	N/A		844-5184	1.53-7.1
Inborn Errors of Immunity (IEI) Patients						
CTLA4 haploinsufficiency (c.380A>G p.(Tyr127Cys)	45	F	Ig Replacement	9.56	1040	7.7
CTLA4 haploinsufficiency (c.380A>G p.(Tyr127Cys)	19	F	Ig Replacement	3.45	1640	12.6
LRBA compound heterozygote (c.1896C>T p.R633X) and (c.2258+4dupA p.V723_K753del)	38	F	Ig Replacement	11.1	4	6.93
NFKB1: c.1423delG, p.(Ala475ProfsTer10)	52	M	Ig Replacement	19.39	4	5.52
NFKB1: c.160-1G>A	22	F	None	12.62	2084	1.86
NFKB1: c.1901dupT p.(Leu636ThrfsTer11)	26	F	Ig Replacement	9.16	4	19
NFKB1c.1423delG, p.(Ala475ProfsTer10)	59	M	None	4.55	138	4.06
NFKB1: c.1190dupG p.(T398H fsTer9)	61	M	None	8.05	3074	4.42
No causative mutation found	23	M	Ig Replacement	1.4	NS	19.3
Immune Checkpoint Blockade (ICB) Treated Patients						
Stage IIIA Oesophageal cancer	55	M	Ipilimumab Nivolumab		1636	8.62
Stage IIIC Melanoma	57	M	Pembrolizumab		300	11
Stage IV Metastatic Melanoma	62	M	Pembrolizumab		554	10.6
Stage IV Metastatic Renal	52	F	Ipilimumab Nivolumab		824	7.17
Stage IV Metastatic Renal	67	M	Ipilimumab* Nivolumab		4	31.9
Stage IIIB Melanoma	63	F	Pembrolizumab		212	7.21
Stage IV Metastatic Melanoma	43	M	Pembrolizumab / Ipilimumab Nivolumab		4736	3.2
Stage IIIC Melanoma	59	M	Pembrolizumab		276	9.57
Stage IIIB Melanoma	59	F	Pembrolizumab		440	6.25
Stage IV Metastatic Melanoma	51	M	Ipilimumab Nivolumab		2488	8.99
Stage IV Metastatic Melanoma	62	M	Pembrolizumab		1602	3.07
Stage IV Metastatic Melanoma	76	M	Pembrolizumab		669.8	11.7
Stage IV Metastatic Melanoma	75	M	Pembrolizumab		256.4	8.73
Stage IV Melanoma	69	M	Nivolumab		2100	5.48
Stage IIIC Melanoma	73	M	Pembrolizumab		813.2	4.98
Stage IV Metastatic Melanoma	75	M	Pembrolizumab		835.2	3.72
Stage IV Melanoma	76	F	Nivolumab		351	19.5
Stage IV Metastatic Renal	70	M	Ipilimumab Nivolumab		NS	22.6
Stage IV Metastatic Melanoma	48	F	Pembrolizumab		464	9.47

IgG, Immunoglobulin G; NT₅₀, Neutralisation titres; ABCs, Aged-associated B cells; NS, No sample collection

*Patient was off of this treatment at the time of vaccination and sampling

(2) The authors report that the ABC frequency is higher in patients with SLE, similarly to a higher frequency found in females amongst healthy controls. SLE is known to have a female dominance, thus raising whether the association is linked to the existence of autoimmune disease itself, or rather it reflects its higher occurrence amongst females. Have the patients within the SLE group also been stratified according to their biological gender?

We thank the reviewer for this this suggestion. The SLE group consisted of only two females and one male patient, limiting our scope to address this question directly. Nonetheless, we have subsetting our single cell data according to sex, and analysed the proportion of classical ABCs across all groups. Compared with same-sex healthy controls, the frequency of ABCs was higher in all three individuals with SLE, although the difference was more marked in the female subjects. This is displayed in Sup. Fig. 1f.

Correspondingly, we have added the following text to the results on page 5:

“Classical ABC frequencies were notably higher in females, and patients with SLE (Fig. 1b-d and Supplementary Fig. 1c-f).”

Importantly, to confirm that the accumulation of ABCs negatively impacts the humoral response independent of sex, we also tested the correlation between neutralising capacity at different times after vaccination with pre-vaccination levels of ABCs in the cohort subsetting by sex. These data are now displayed in Sup. Fig. 8a:

Correspondingly, we have added the following text to the results on page 8:

“A similar negative correlation between ABC frequency and neutralising capacity was still observed when we subdivided the cohort according to sex (Supplementary Fig. 8a)”

(3) Throughout the manuscript the numeric parameters for B cells/ABCs and their subsets are expressed as frequency; given that the frequency of B cells is higher, while the frequency of ABCs is lower in healthy individuals [Fig. 3c] (and the opposite is found in both immune deficient and ICB patients), yet can the absolute number of ABCs/unit of blood be similar in all three groups?

We thank the reviewer for making this point. Unfortunately, the study was conducted during a time of national lockdown, and we were unable to obtain TruCount tubes for enumeration of absolute cell counts. We have noted this in the new statement of study limitations on page 11:

“Third, absolute numbers of ABCs were not quantified in this study, it is not possible to ascertain whether the number of ABCs or the skewing of the B cell repertoire towards this differentiation state, leads to the impairment in vaccine responses.”

(4) In Fig 4a/b, the authors claim that "Strikingly, RBD-binding B cells were always significantly less frequent in patients than in healthy controls"; one panel of the graphs shows actually a higher frequency of Ag-specific B cells in ICB patient on D21 [0,042 vs. 0,054] (in panel a), and there is no indication of significance (in panel b) supporting the claim. Furthermore, the fold-increase of these cells in both the IEI and ICB groups actually seems higher than in the healthy controls - please comment.

We apologise for the wording of this statement, which has inadvertently misled the reviewer. By “always”, we meant “in every group”, not “in every individual”. We have therefore adjust the text on page 8 accordingly:

“Strikingly, RBD-binding B cells were significantly less frequent in the different patient groups than in healthy controls (**Fig. 4a-b**).”

In addition, we have also now picked a FACS plot which is more representative of that cohort, and displayed in Fig. 4a:

Finally, we thank the reviewer for pointing out the absence of p values from that figure. We have now updated Fig. 4b accordingly

In respect of the fold-increases in these cells in response to vaccination, contrary to the visual impression, we did not observe significant differences between cases and controls.

We have not shown these derived data in the manuscript, but include them below (and would be happy to do so if the reviewer or editor requests):

(5) Were there any attempts to perform in vitro stimulation tests on B cells, to compare various groups for investigating the degree of autonomous B-cell intrinsic preference for ABC formation? This would be important to support the claim in conclusion on the necessity for repeated Covid vaccination in patients with ICB, lest the immunization elicits a disproportionately larger number of ABCs with dysfunctional features, that would compromise the efficiency of repeated vaccination.

We thank the reviewer for making this point. Since we have not performed in vitro differentiation tests on B cells, we agree that it is prudent to remove the comments about repeated COVID-19 vaccination in patients treated with ICB from the discussion.

Reviewer #3 (Remarks to the Author)

The manuscript by Yam-Puc et al. compared the transcriptional profile of age-associated B cells (ABCs) across cohorts of healthy controls, cancer patients receiving immune checkpoint blockade (ICB) therapy and patients with NFkB or CTLA-4 haploinsufficiency (IEI). They found that ABCs were transcriptionally similar across the different cohorts but elevated in ICB and IEI patients. They found that the ABCs expressed AIRE and displayed an increased profile of tissue-restricted antigens, which is typically restricted to the thymus. The increased frequency of ABCs correlated with decreased response to Covid19 vaccinations, including decreased maintenance of neutralizing antibodies.

The observation that ABCs display a similar transcriptional profile between the healthy controls and ICB and IEI patients is interesting. However, the subsequent figures in the manuscript rely completely on correlations, drawing conclusions of the role of ABCs in vaccine induced Ab production that is not supported by any functional assays.

We thank the reviewer for raising this point and in response we now present experiments that support our transcriptional data. In these assays we assess immune complex binding and cytokine secretion capacity in healthy controls and patients. In both sets of assays we identify similar functional capacities of ABCs irrespective of the cause for their expansion. Furthermore, our finding that ABCs express the inhibitory receptor FcγRIIB and exhibit greater Fc dependent immune complex binding presents two mechanisms by which increased ABCs might diminish humoral vaccine responses.

This data is displayed in Fig. 5 below.

and described in the text on page 10 as follows:

“Assessment of ABCs for immune complex binding and cytokine secretion

Preclinical evidence has suggested that ABCs might impede humoral vaccine responses by diminishing affinity maturation⁵¹. Affinity maturation is a process requiring iterative selection of proliferating B cell clones within the germinal centre, and this takes time. However, we found substantial neutralising capacity within 8 days of vaccination, suggesting a limited requirement for substantial further affinity maturation. Furthermore, the negative correlation between ABC frequency and neutralising capacity was greatest at early time points after repeat immunisation. We therefore investigated other mechanisms by which ABCs could limit humoral vaccine responses. Further scRNAseq analysis indicated that ABCs, particularly classical ABCs, expressed high levels of the inhibitory Fc gamma receptor IIB (Fc γ RIIB or CD32B) (Fig. 5a). Expression of other Fc receptors was not observed (Supplementary Fig. 12a), consistent with the literature suggesting this is the only Fc receptor expressed on B lymphocytes⁵². To test this functionally, we generated immune complexes by incubation of fluorescent rabbit anti-human IgG with polyclonal human IgG at a ratio of 1:2. This resulted in complexes three times the molecular weight of rabbit IgG, consistent with a trimolecular antibody stoichiometry (Supplementary Fig. 12b). The binding of these complexes to B cells was Fc-dependent, and significantly increased in ABCs from all patient cohorts relative to other B cell subsets

(Fig. 5b and 5d). This positions ABCs as the B cell subset best placed to clear immune complexes directed against vaccine antigen, potentially reducing the longevity of antigen availability. Furthermore, the differentiation of RBD-specific ABCs may be inhibited by signalling through the FcγRIIB, limiting their contribution to memory B cell expansion or antibody secreting cell differentiation. Production of cytokines has been suggested as one mechanism by which ABCs limit B cell function^{53,54} and in preclinical models of NFKB1 deficiency B cell production of IL-6 contributes to disease pathogenesis⁵⁵. We therefore evaluated cytokine production capacity by in-vitro stimulation of B cells subsets with PMA/ionomycin and evaluated intracellular cytokines production by FACS (Fig. 5c and Supplementary Fig. 12c). This showed that although ABCs from all patient groups can produce IL-6 and TNF-α (Fig. 5e), the frequency of these was not higher than in other B cell subsets. Taken together within our transcriptional assessment, these results indicate ABCs from different patient groups are functionally similar and present additional mechanisms by which elevated ABCs may diminish humoral vaccine responses.”

There are likely a number of other factors that can impact maintenance of the antibody response in these patients as they display either genetically, or therapy, induced alterations in their broader immune response, which will include CD4 T cells that help in antibody production.

We thank the reviewer for raising this important point. We have addressed this in the manuscript in a number of ways. First, we have now evaluated the T cell responses across the different groups, and presented these data in Sup. Fig. 11:

Correspondingly, we have added the following text to the results on page 9:

“Humoral vaccine responses require B and T lymphocyte interaction, we next assessed T cells. Circulating T follicular helper cell (Tfh) frequencies were actually increased in patients with IEI and unchanged in patients treated with ICB (Supplementary Fig. 11a-b). Similarly, activated (OX40+CD137+) CD4 T cell frequency was increased in patients with IEI before vaccination and this was maintained at day 21. No significant difference was observed in CD8+ T cell activation markers, nor in patients treated with ICB (Supplementary Fig. 11c). To assess antigen-specific responses, we next enumerated Spike-specific T cells using IFN- γ ELISpot (Supplementary Fig. 11d). No significant differences were observed between healthy controls and patient groups. Finally, we assessed the correlation between neutralising capacity and pre-vaccination levels of circulating Tfh, but found no significant association. Taken together, these results suggest that the association between increased ABC frequency and diminished humoral vaccine response is independent of T cell function.”

Second, we have examined some key co-variables which could affect the immune response in patients with cancer, and presented these data in Sup. Fig. 10:

Correspondingly, we have added the following text to the results on page 9:

“Among patients treated with ICB, we observed similar RBD-specific B cell frequencies and neutralising responses when stratified by steroidal treatment, dual/single checkpoint blockade therapy and cancer stage (III/IV) (Supplementary Fig. 10a-b). Conversely, we noted a reduction in both RBD specific B cell frequency and the neutralising antibody response with increasing age (Supplementary Fig. 10c). In theory, several factors (such as comorbidity) could impede the vaccination responses of older individuals with cancer, independent of ABC frequency. We therefore specifically examined the associations between ABC frequency, age and diminished humoral response amongst individuals under the age of 60. In this subset, RBD-specific B cell frequency and neutralising capacity were still negatively correlated with ABC frequency, but not with age (Supplementary Fig. 10d). These results support a primary association between pre-vaccine ABC frequency and impairment of the humoral response.”

Finally, we have now evaluated some key cytokines relating to the innate response, and presented these data in Sup. Fig. 6:

Correspondingly, we have added the following text to the results on page 7:

“We assessed serum samples 24h after the second dose of BNT162b2 and did not observe any difference in the levels of IL-1β, IL-12 and IFN-α between patients and controls at this time point, suggesting a similar innate immune response to the vaccine (Supplementary Fig. 6).”

Overall, we agree that a number of other factors may impact the maintenance of the antibody response, and now address this in a new section on study limitations in the discussion on page 11:

“There are several limitations of this study. First, the number of people studied was modest, limiting the power to detect small differences in some comparisons and precluding a formal multivariate analysis. Second, many characteristics of ABCs may contribute to an impaired humoral vaccine response, including (but not limited to) reduced affinity maturation, signalling through inhibitory FcγRIIB and clearance of antigen. We have not quantified the extent of affinity maturation in this study, nor yet performed the detailed preclinical experiments necessary to determine the relative contribution of these mechanisms.”

Minor comments:

The abstract sounds like 2 different studies rather than looking at ICB on B cells and confirming phenotypes in vaccinated patients.

We have attempted to address it by re-wording the abstract as follows:

“Age-associated B cells (ABCs) accumulate with age, as well as in individuals with a range of immunological dyscrasias. These include patients with cancer treated with immune checkpoint blockade and patients with inborn errors of immunity. In this study, we sought to determine whether ABCs found in all these conditions are similar, and whether they enhance or detract from the response to COVID-19 vaccination. We use single cell RNA sequencing to show that ABCs arising from distinct aetiologies have common transcriptional profiles and may be subdivided according to the expression of genes associated with different immune functions, such as the autoimmune regulator (AIRE). Next, we perform detailed longitudinal profiling of the COVID-19 vaccination response in patients and controls. We show that high pre-vaccination ABC frequency correlates with decreased levels of antigen-specific memory B cells, and reduced magnitude and longevity of neutralising capacity against SARS-CoV-2 virus. Potentially contributing to this, ABCs express high levels of the inhibitory FcγRIIB receptor and are distinctive in their ability to bind immune complexes. This could contribute to diminished vaccine responses either directly as result of inhibitory signalling or indirectly via enhanced clearance of immune complexed-antigen. Expansion of ABCs may therefore serve as a biomarker identifying individuals at risk of a suboptimal response to COVID-19 vaccination.”

Figure 2d, a better definition of what the tissue restricted antigen score is. At least a supplemental table and expression pattern.

We thank the reviewer for making this point. Accordingly, we have improved the description of the tissue restricted antigen score in the methods section as follows:

“Geneset Scores

A list of 74 Aire-induced genes were obtained from Yamano et al., [14] after converting mouse gene IDs to human gene IDs (Supplementary Table 4). For calculating the gene set score for each cell, the AddModuleScore function from Seurat package was used. Briefly, average expression of genes in each cell is calculated and is subtracted by the aggregated expression of control gene sets. For selecting control gene sets, genes are first binned. Then, for each gene, control gene sets are randomly chosen from the same expression bin as that gene, so that they have similar expression patterns to Aire target genes. To explore the expression pattern of these genes in different tissues, normalised transcript per million (nTPM) values were extracted in different immune cells and different tissues from the human protein atlas website (https://v22.proteinatlas.org/download/rna_immune_cell.tsv.zip and https://v22.proteinatlas.org/download/rna_tissue_consensus.tsv.zip, respectively). In addition to two groups of B lymphocytes, those tissues and cells that have maximum expression of at least one of the genes from our list were kept.”

We have also now included a table of AIRE target genes displayed as Sup. Table 4:

Table S4. AIRE target genes related to Fig. 2d and Supp. Fig. 5

1	CISH	20	MAP3K15	39	ATP2B4	57	TET2
2	GBP7	21	TP73	40	SEMA6D	58	ARHGAP19
3	PVR	22	TFCP2L1	41	SIGLEC6	59	CD3E
4	DDX60	23	INSYN2B	42	HLA-G	60	CD8A
5	PARP12	24	FNIP1	43	CDH17	61	RAG1
6	LAMP3	25	RAPGEF4	44	VWA3B	62	NLRCS
7	HLA-F	26	SYN3	45	GGN	63	SLFN12L
8	VASN	27	GRIK2	46	ZBTB32	64	ZBP1
9	SOCS3	28	ACSBG1	47	NEK2	65	PHF11
10	LIF	29	HSPH1	48	GAS2L3	66	OAS2
11	AHR	30	EML5	49	TDG	67	MX1
12	HK2	31	ST8SIA1	50	FLT3	68	CASTOR1
13	SLC24A1	32	ZBTB18	51	SCIMP	69	IL2RB
14	STXBP1	33	DBNDD1	52	PIK3R6	70	CTSW
15	STARD9	34	RANBP17	53	ADGRG5	71	RUNX2
16	OBSCN	35	ATP10A	54	RSAD2	72	SLFN12
17	SSPN	36	SEMA6A	55	HLA-E	73	CSF2RB
18	STRIP2	37	RPRM	56	STK38L	74	SOCS2
19	SLC5A3	38	PKD2				

Finally, we have calculated the tissue-specific expression pattern for each gene, and display it as Sup. Fig. 5:

This is described in the results on page 6:

“AIRE-expressing cells also upregulated a set of genes previously defined as AIRE targets in thymic B cells⁴¹, which are predominantly expressed in other tissues such as brain (Fig. 2d, Supplementary Table 4 and Supplementary Fig. 5).”

Reviewer #4 (Immune cell biology, transcriptome analyses) (Remarks to the Author):

In this manuscript, Yam-Puc et. al. studies the effect of age-associated B cells and the response to covid vaccination. Their study groups include healthy control, cancer patients treated with immune checkpoint blockade, patients with NFKB1 and CTLA4 haploinsufficiency, and SLE. They identified 1) common transcriptional profiles for ABCs in the study groups, 2) distinct ABCs clusters between groups, 3) classical ABCs express AIRE, and 4) ABCs frequency predicts neutralizing Ab responses to COVID vaccination. Based on these observations, the authors conclude that ABC frequency might be a predictive biomarker for reduced vaccine protection. The data is interesting, but mostly descriptive and don't provide any mechanistic explanation for the observed results.

We agree that uncovering a mechanistic explanation for our observations will be critical. To do so will ultimately require a further substantive body of work. Nonetheless, we have now included further analyses and new experiments which directly address two leading hypotheses. In brief, we have found that ABCs have the capacity to produce key cytokines. Equally, we discovered that ABCs express high levels of *FCGR2B*, and that this is associated with an increased ability to bind immune complexes. These data are shown in a new figure (Fig. 5):

Correspondingly, we have added the following text to the results on page 10:

“Assessment of ABCs for immune complex binding and cytokine secretion

Preclinical evidence has suggested that ABCs might impede humoral vaccine responses by diminishing affinity maturation⁵¹. Affinity maturation is a process requiring iterative selection of proliferating B cell clones within the germinal centre, and this takes time. However, we found substantial neutralising capacity within 8 days of vaccination, suggesting a limited requirement for substantial further affinity maturation. Furthermore, the negative correlation between ABC frequency and neutralising capacity was greatest at early time points after repeat immunisation. We therefore investigated other mechanisms by which ABCs could limit humoral vaccine responses. Further scRNAseq analysis indicated that ABCs, particularly classical ABCs, expressed high levels of the inhibitory Fc gamma receptor IIB (FcγRIIB or CD32B) (Fig. 5a). Expression of other Fc receptors was not observed (Supplementary Fig. 12a), consistent with the literature suggesting this is the only Fc receptor expressed on B lymphocytes⁵². To test this functionally, we generated immune complexes by incubation of fluorescent rabbit anti-human IgG with polyclonal human IgG at a ratio of 1:2. This resulted in complexes three times the molecular weight of rabbit IgG, consistent with a trimolecular antibody stoichiometry (Supplementary Fig. 12b). The binding of these complexes to B cells was Fc-dependent, and significantly increased in ABCs from all patient cohorts relative to other B cell subsets (Fig. 5b and 5d). This positions ABCs as the B cell subset best placed to clear immune complexes directed against vaccine antigen, potentially reducing the longevity of antigen availability. Furthermore, the differentiation of RBD-specific ABCs may be inhibited by signalling through the FcγRIIB, limiting their contribution to memory B cell expansion or antibody secreting cell differentiation. Production of cytokines has been suggested as one mechanism by which ABCs limit B cell function^{53,54} and in preclinical models of NFKB1 deficiency B cell production of IL-6 contributes to disease pathogenesis⁵⁵. We therefore evaluated cytokine production capacity by in-vitro stimulation of B cells subsets with PMA/ionomycin and evaluated intracellular cytokines production by FACS (Fig. 5c and Supplementary Fig. 12c). This showed that although ABCs from all patient groups can produce IL-6 and TNF-α (Fig. 5e), the frequency of these was not higher than in other B cell subsets. Taken together within our transcriptional assessment, these results indicate ABCs from different patient groups are functionally similar and present additional mechanisms by which elevated ABCs may diminish humoral vaccine responses.”

In addition, we have including the following in a new statement of study limitations in page 11 of the discussion:

“Second, many characteristics of ABCs may contribute to an impaired humoral vaccine response, including (but not limited to) reduced affinity maturation, signalling through inhibitory FcγRIIB and clearance of antigen. We have not quantified the extent of affinity maturation in this study, nor yet performed the detailed preclinical experiments necessary to determine the relative contribution of these mechanisms.”

Comment #1: Also, the sample size is small. Most of the observed findings need to be confirmed in a larger N. For example, in figure 3c, about 40% of IEI and 50% of ICB subjects has similar % of B cells compared to HC, but the authors shoed some significant differences

between HC and IEI. Similar observation is also true for % of ABCs graph. Only few patients have higher frequency of ABCs. Thus, more N is needed to confirm these results.

We thank the reviewer for this point. Unfortunately, the timing of this study was dictated by the COVID-19 pandemic, and associated vaccination campaign – and it is not possible now to go back and collect more patient samples. For maximum transparency, we have (where appropriate) already shown individual data points in all figures, and performed appropriate statistical tests of significance for all highlighted comparisons.

In respect of the reviewer’s specific point (the extent of the overlap between groups), we have re-analysed the correlations between ABC frequency and neutralising capacity for the subset of subjects with ABC or B cell frequencies falling within the range seen for healthy controls. These data are displayed in Sup. Fig. 8b-c:

Correspondingly, we have added the following text to the results on page 8:

“A similar negative correlation between ABC frequency and neutralising capacity was still observed when we subdivided the cohort according to sex (Supplementary Fig. 8a), when the analysis was restricted to those subjects with ABC frequencies falling within the range observed in healthy controls (Supplementary Fig. 8b), and when the analysis was restricted to those subjects with B cell frequencies falling within the range of healthy controls (Supplementary Fig. 8c).”

In addition, we have specifically addressed this in the new section on study limitations in the discussion on page 11:

“There are several limitations of this study. First, the number of people studied was modest, limiting the power to detect small differences in some comparisons and precluding a formal multivariate analysis.”

Comment #2: the identification of ABCs is not optimal, as ABCs are commonly identified by the expression of T-bet, CD11c, CD11b, and lack of CD21. It is not clear why the authors only used CD11c and CD21 as markers to define ABCs?

We thank the reviewer for raising this point. There is a number of different markers used in the extensive literature describing ABCs. Initially many of the characteristics of ABCs could be defined in the population of B cells that has reduced CD21 expression⁸⁻¹⁰. Although T-bet is the transcription factor most associated with the ABC differentiation state, recent data from inherited immunodeficiency studies demonstrates that CD11c expression depends on T-bet¹¹. However, since our major conclusion relates to a potential biomarker we wanted to present a relatively simple FACS panel that can be readily incorporated into diagnostic laboratory assessments. We have modified the text on page 5 to describe this as follows:

“Using CITE-seq, we identified ABCs that had low CD21 and high CD11c surface protein expression, in keeping with the definitions used in some previous flow cytometry-based studies^{16,34} (Supplementary Fig. 1a-b).”

Comment #3: In figure 2a, it's not clear why the authors only focused on AIRE gene induction? There were many other genes that were upregulated in ABCs. The authors need to at least discuss the relevance of some of other differentially expressed/upregulated genes.

We thank the reviewer for making this point. We were of course intrigued by the upregulation of AIRE because of its known involvement in the expression of tissue-restricted antigens, and the association between ABCs and autoimmunity¹². We have added the following text to the results on page 6 (the potential relevance of AIRE is already discussed later in the paragraph):

“Consistent with our gene ontology analysis, we confirmed MHC-II antigen presentation related genes, such as HLA-proteins, CD74 and CD86, were amongst the most differentially expressed genes of Classical ABCs. Additionally, other important genes involved in the processing of peptides, oligosaccharides and fatty acids such as LGMN, IFI30, PSAP and ASAH1 were differentially expressed (Fig. 2a and Supplementary Table 3). Intriguingly, this analysis also revealed the Autoimmune Regulator (AIRE) to be significantly upregulated in Classical ABCs from all subject cohorts (Fig. 2a-b, Supplementary Fig. 4).”

Comment #4: In figure 4a-b, the author showed some minor differences in ABCs between HC and patients, but it is necessary to also show functional differences such as production of IFN γ , TNF, IL-17, IL-10, or IL-4 cytokines or other readouts. Is there any functional differences in cytokine profile between vaccinated and unvaccinated subject? And does the disease state affects this response?

We thank the reviewer for raising this important issue, which we have addressed in response to their summary comments (as above). In brief, whilst not impacting the validity of ABCs as a biomarker identifying individuals at risk of a suboptimal response to COVID-19 vaccination, we agree that uncovering a mechanistic explanation for our observations will be critical. To do so will ultimately require a further substantive body of work, which is beyond the scope of this study. Nonetheless, we have now included further analyses and new experiments which directly address two leading hypotheses, including the production of key cytokines. Finally, we have examined some key co-variables which could affect the immune response in patients with cancer, and presented these data in Sup. Fig. 10:

Correspondingly, we have added the following text to the results on page 9:

“Among patients treated with ICB, we observed similar RBD-specific B cell frequencies and neutralising responses when stratified by steroidal treatment, dual/single checkpoint blockade therapy and cancer stage (III/IV) (Supplementary Fig. 10a-b). Conversely, we noted a reduction in both RBD specific B cell frequency and the neutralising antibody response with increasing age (Supplementary Fig. 10c). In

theory, several factors (such as comorbidity) could impede the vaccination responses of older individuals with cancer, independent of ABC frequency. We therefore specifically examined the associations between ABC frequency, age and diminished humoral response amongst individuals under the age of 60. In this subset, RBD-specific B cell frequency and neutralising capacity were still negatively correlated with ABC frequency, but not with age (Supplementary Fig. 10d). These results support a primary association between pre-vaccine ABC frequency and impairment of the humoral response.”

Additional references for responses to reviewers’ comments:

- 1 Khoury, D. S. *et al.* Neutralizing antibody levels are highly predictive of immune protection from symptomatic SARS-CoV-2 infection. *Nat Med* **27**, 1205-1211, doi:10.1038/s41591-021-01377-8 (2021).
- 2 van der Klaauw, A. A. *et al.* Accelerated waning of the humoral response to SARS-CoV-2 vaccines in obesity. *medRxiv*, 2022.2006.2009.22276196, doi:10.1101/2022.06.09.22276196 (2022).
- 3 Malik, P. *et al.* Biomarkers and outcomes of COVID-19 hospitalisations: systematic review and meta-analysis. *BMJ Evid Based Med* **26**, 107-108, doi:10.1136/bmjebm-2020-111536 (2021).
- 4 Lai, Y. J. *et al.* Biomarkers in long COVID-19: A systematic review. *Front Med (Lausanne)* **10**, 1085988, doi:10.3389/fmed.2023.1085988 (2023).
- 5 Samprathi, M. & Jayashree, M. Biomarkers in COVID-19: An Up-To-Date Review. *Front Pediatr* **8**, 607647, doi:10.3389/fped.2020.607647 (2020).
- 6 Zhang, W. *et al.* Excessive CD11c(+)Tbet(+) B cells promote aberrant TFH differentiation and affinity-based germinal center selection in lupus. *Proc Natl Acad Sci U S A* **116**, 18550-18560, doi:10.1073/pnas.1901340116 (2019).
- 7 Collier, D. A. *et al.* Age-related immune response heterogeneity to SARS-CoV-2 vaccine BNT162b2. *Nature* **596**, 417-422, doi:10.1038/s41586-021-03739-1 (2021).
- 8 Warnatz, K. *et al.* Expansion of CD19(hi)CD21(lo/neg) B cells in common variable immunodeficiency (CVID) patients with autoimmune cytopenia. *Immunobiology* **206**, 502-513, doi:10.1078/0171-2985-00198 (2002).
- 9 Wehr, C. *et al.* A new CD21low B cell population in the peripheral blood of patients with SLE. *Clin Immunol* **113**, 161-171, doi:10.1016/j.clim.2004.05.010 (2004).
- 10 Rakhmanov, M. *et al.* Circulating CD21low B cells in common variable immunodeficiency resemble tissue homing, innate-like B cells. *Proc Natl Acad Sci U S A* **106**, 13451-13456, doi:10.1073/pnas.0901984106 (2009).
- 11 Yang, R. *et al.* Human T-bet governs the generation of a distinct subset of CD11c(high)CD21(low) B cells. *Sci Immunol* **7**, eabq3277, doi:10.1126/sciimmunol.abq3277 (2022).
- 12 Cancro, M. P. Age-Associated B Cells. *Annu Rev Immunol* **38**, 315-340, doi:10.1146/annurev-immunol-092419-031130 (2020).

REVIEWERS' COMMENTS

Reviewer #1 (Remarks to the Author):

The reviewers note that there remain mechanistic gaps although major concerns were addressed, and rebuttal of the authors is quite extensive. Notably, there are now several subtypes of ABC with differences in phenotype and gene expression identified. There remain substantial work to do.

On another topic the authors might consider adding qualifiers to their sentence on page 8, line 222.

"Whilst circulating antibodies derived from plasma cells wane over time, long-lived immunological memory can persist in expanded clones of antigen-specific memory B cells."

Anti-tetanus plasma cells can persist many decades, and may be the best example of "long-lived plasma cells. Radbruch and others have shown these long-lived plasma cells in the bone marrow do not require a high level of homeostatic replacement.

Should the sentence on page 8 be revised or clarified to better explain the intended meaning, which presumably meant to focus on the antibody products of ABC?

Reviewer #2 (Remarks to the Author):

In this revised manuscript by Thaventhiran et al. my previous comments and queries have been adequately addressed. No further critiques from this reviewer.

Reviewer #3 (Remarks to the Author):

I appreciate the amount of work the authors have added to the manuscript and revisions made to the text. The story is much more engaging and the functional data supports the original conclusions. My comments were addressed. I have no further comments.

Reviewer #4 (Remarks to the Author):

All of my concerns have been adequately addressed by the authors. The edited manuscript is significantly improved.

REVIEWERS' COMMENTS

Reviewer #1 (Remarks to the Author):

The reviewers note that there remain mechanistic gaps although major concerns were addressed, and rebuttal of the authors is quite extensive. Notably, there are now several subtypes of ABC with differences in phenotype and gene expression identified. There remain substantial work to do. On another topic the authors might consider adding qualifiers to their sentence on page 8, line 222.

“Whilst circulating antibodies derived from plasma cells wane over time, long-lived immunological memory can persist in expanded clones of antigen-specific memory B cells.” Anti-tetanus plasma cells can persist many decades, and may be the best example of “long-lived plasma cells. Radbruch and others have shown these long-lived plasma cells in the bone marrow do not require a high level of homeostatic replacement. Should the sentence on page 8 be revised or clarified to better explain the intended meaning, which presumably meant to focus on the antibody products of ABC?

We thank the reviewer for suggesting we address this point. We have revised the sentence to clarify it. Correspondingly, we have added the following change to the text on page 8:

“In addition to circulating antibodies derived from plasma cells, long-lived immunological memory can persist in expanded clones of antigen-specific memory B cells, both arms of B cell responses can better counteract the same pathogen during subsequent encounters.”

Reviewer #2 (Remarks to the Author):

In this revised manuscript by Thaventhiran et al. my previous comments and queries have been adequately addressed. No further critiques from this reviewer.

We thank the reviewer for all their suggestions and comments. These have helped us to greatly improve our manuscript.

Reviewer #3 (Remarks to the Author):

I appreciate the amount of work the authors have added to the manuscript and revisions made to the text. The story is much more engaging and the functional data supports the original conclusions. My comments were addressed. I have no further comments.

We thank the reviewer for all their suggestions and comments. These have helped us to greatly improve our manuscript.

Reviewer #4 (Remarks to the Author):

All of my concerns have been adequately addressed by the authors. The edited manuscript is significantly improved.

We thank the reviewer for all their suggestions and comments. These have helped us to greatly improve our manuscript.